



# Hydrography, transport and mixing of the West Spitsbergen Current: the Svalbard Branch in summer 2015

Eivind Kolås[1] and Ilker Fer[1]

[1]Geophysical Institute, University of Bergen, Bergen, Norway

*Correspondence to:* Ilker Fer (ilker.fer@uib.no)

**Abstract.** Measurements of ocean currents, stratification and microstructure were made in August 2015, northwest of Svalbard, downstream of the Atlantic inflow in Fram Strait in the Arctic Ocean. Observations in three sections are used to characterize the evolution of the West Spitsbergen Current (WSC) along a 170-km downstream distance. Two alternative calculations imply 1.5 to 2 Sv (1 Sv = $10^6$ m$^3$ s$^{-1}$) is routed to recirculation and Yermak branch in Fram Strait, whereas 0.6 to 1.3 Sv is carried by the Svalbard branch. The WSC cools at a rate of 0.20 °C per 100 km, with associated bulk heat loss per along-path meter of $(1.1\text{-}1.4)\times10^7$ W m$^{-1}$, corresponding to a surface heat loss of 380-550 W m$^{-2}$. The measured turbulent heat flux is too small to account for this cooling rate. Estimates using a plausible range of parameters suggest that the contribution of diffusion by eddies could be limited to one half of the observed heat loss. In addition to shear-driven mixing beneath the WSC core, we observe energetic convective mixing of an unstable bottom boundary layer on the slope, driven by Ekman advection of buoyant water across the slope. The estimated lateral buoyancy flux is $O(10^{-8})$ W kg$^{-1}$, sufficient to maintain a large fraction of the observed dissipation rates, and corresponds to a heat flux of approximately 400 W m$^{-2}$. Convectively-driven bottom mixing followed by the detachment of the mixed fluid, and its transfer into the ocean interior can lead to substantial cooling of the WSC, at a rate comparable to that expected from diffusion by eddies.

## 1 Introduction

The Arctic Ocean contributes to the global ocean thermohaline circulation through exchanges in Fram Strait, which is the main connection to the Atlantic Ocean (Aagaard et al., 1985). The total volume transport through Fram Strait is $9 \pm 2$ Sv (1 Sv = $10^6$ m$^3$ s$^{-1}$) northward, and $12 \pm 1$ Sv southward (Fahrbach et al., 2001; Schauer et al., 2004). A large fraction of the northward flow is the West Spitsbergen Current (WSC), located on the eastern side of Fram Strait, which is a northward flowing extension of the Norwegian Atlantic Current. The mean net volume transport in the WSC, measured along an array at 78°50′N in the period between 1997 and 2010, is $6.6 \pm 0.4$ Sv (Beszczynska-Möller et al., 2012), of which $3.0 \pm 0.2$ Sv is Atlantic Water (AW) with temperature above 2°C. The WSC continues as a topographically-guided boundary current, contributing to the circumpolar boundary current downstream. Between Fram Strait and the Lomonosov Ridge, the boundary current slows down from about 0.25 m s$^{-1}$ to 0.06 m s$^{-1}$, and changes structure from a mainly barotropic flow to a baroclinic flow (Pnyushkov et al., 2015). AW transported by the WSC is the major heat and salinity source for the Arctic Ocean (Boyd and D'Asaro, 1994;



Aagaard et al., 1985; Rudels et al., 2015), and Arctic conditions are highly influenced by changes in the AW inflow properties (Polyakov et al., 2017).

The circulation of AW in Fram Strait has multiple branches (Fig. 1a). The WSC flows at a steady pace of approximately 0.25 m s$^{-1}$, along the 1000-m isobath, from Bear Island at 74°30′N to the southern flanks of the Yermak Plateau (YP) at
79°30′N (Boyd and D'Asaro, 1994). Observations show that the WSC splits into two branches where the isobaths diverge near the YP, an outer branch following the 1000-m isobath, and an inner branch (the Svalbard branch) following the 400-m isobath (Aagaard et al., 1987; Farrelly et al., 1985; Cokelet et al., 2008). The Svalbard branch has a 40-km wide core with a strong barotropic component, and flows approximately along the $f/H$ contours around Svalbard (Aagaard et al., 1987; Perkin and Lewis, 1984) ($f$ is the Coriolis parameter and $H$ is water depth). The outer branch, however, has a 60-km wide core and does
not follow the 1000-m isobath as closely (Aagaard et al., 1987). Observations suggest that the outer branch splits into three different branches. A part of the flow detaches from the 1000-m isobath and recirculates in Fram Strait, contributing with warm and salty water to the southward flow on the Greenland slope (Aagaard et al., 1987; Farrelly et al., 1985; Beszczynska-Möller et al., 2012; Hattermann et al., 2016). The main recirculation is on the northern rim of the Molloy Hole at approximately 80°N and 4°E (Hattermann et al., 2016). The remaining part of the outer branch following the 1000-m isobath is called the Yermak
branch, and flows along the outer flanks of the YP, possibly rejoining the Svalbard branch where the isobaths converge north of Svalbard at approximately 80°30′N and 13°E (Perkin and Lewis, 1984; Cokelet et al., 2008; Våge et al., 2016; Meyer et al., 2016). Acoustically-tracked subsurface floats revealed a shortcut across the YP (Gascard et al., 1995), through a topographic passage at 80°45′N and 6°E. The presence of the Yermak Pass branch is supported by numerical model results, showing flow of AW following the 700-800-m isobaths before rejoining the Svalbard branch (Koenig et al., 2017).

The outer branch was reported to contain eddies with diameters of approximately 20 km, that are shed where the two branches split (Padman and Dillon, 1991; Perkin and Lewis, 1984). These eddies control the amount of the AW recirculation in Fram Strait, and therefore, play a major role in the salt and heat budget of the Arctic Ocean (von Appen et al., 2016; Hattermann et al., 2016).

As the AW flows toward the Arctic Ocean, its salinity and temperature properties change as a result of interactions with
the atmosphere and sea ice, and mixing with the surrounding waters. Notable studies reporting the observed property changes of WSC are from observations from a winter cruise in January-February 1989 (Boyd and D'Asaro, 1994), from a fall cruise in October-November 2001 (Cokelet et al., 2008), and from an analysis of a 50-year hydrography data set (1949–1999), reported for summer (August–October) and winter (March–May) seasons (Saloranta and Haugan, 2004). Estimates of along-path freshening of the WSC, measured in the practical salinity scale, are 0.013/100 km in fall 2001 (Cokelet et al., 2008), and
0.010/100 km in summer (Saloranta and Haugan, 2004). The summer/fall cooling rate is 0.19°C/100 km (fall 2001, Cokelet et al., 2008) and 0.20°C/100 km (50-year summer-mean, Saloranta and Haugan, 2004). The cooling rate in winter is 0.4-0.5°C/100 km (winter 1989, Boyd and D'Asaro, 1994), and 0.31°C/100 km (50-year winter-mean, Saloranta and Haugan, 2004). Assuming an AW layer between 100 m and 500 m depth, the cooling rate is equivalent to a heat loss of 310-330 W m$^{-2}$ in summer (Aagaard et al., 1987; Saloranta and Haugan, 2004; Cokelet et al., 2008) and 1050 W m$^{-2}$ in winter (Saloranta and





Haugan, 2004). The cooling of the WSC stream tube observed in winter 1989 implies approximately 900 W m$^{-2}$, limited to a 22-km wide core (Boyd and D'Asaro, 1994).

Boyd and D'Asaro (1994) describe the cooling of the WSC in winter as a three-stage process: Cooling by the atmosphere, cooling by sea ice and cooling by eddy driven mixing along isopycnals. The relative role of the different cooling processes is not

clear. Numerical linear stability analyses using idealized current profile and topography suggest that heat loss contribution from isopycnal diffusion as a result of barotropic instability corresponds to an along-shelf cooling rate of 0.08°C/100 km (Teigen et al., 2010). Extension of the analysis to a two-layer model shows that the baroclinic instability occurs, most pronounced during winter/spring, leading to a heat loss reaching 240 W m$^{-2}$, from the core of the WSC to atmosphere (Teigen et al., 2011).

All these studies agree that vertical mixing alone cannot account for the observed cooling rates. Fer et al. (2010) conclude that internal-wave activity and mixing show variability related to topography and hydrography; thus, the path of the WSC will affect the cooling and freshening rates the AW experiences. Over the steep slopes and prominent topography of the YP, and over the core of the AW branches, vertical mixing can play an important role in modifying the AW properties (Padman and Dillon, 1991; Meyer et al., 2016; Sirevaag and Fer, 2009). In the surface mixed layer in proximity to the WSC, turbulent heat

fluxes of 200-300 W m$^{-2}$ were measured (Sirevaag and Fer, 2009). Once the AW subducts, the vertical mixing is suppressed by overlaying strong stratification, reducing the heat loss to the atmosphere. Padman and Dillon (1991) found that the upward heat flux from the Atlantic layer over the YP slope was 25 W m$^{-2}$, and of that about 6 W m$^{-2}$ entered the mixed layer. At the core of the Svalbard branch, Fer et al. (2010) observed that near-bottom mixing removes 15 W m$^{-2}$ from the AW layer. Outside the WSC, near the northeastern flank of the YP, Sirevaag and Fer (2009) found an average vertical heat flux of 2 W

m$^{-2}$, comparable to the annual oceanic heat flux of 3-4 W m$^{-2}$ to the Arctic pack ice as reported by Krishfield and Perovich (2005).

Here we report summer observations of ocean stratification, currents and microstructure from north of Svalbard near the YP, collected during a cruise in August 2015. Using three sections across the WSC, we present the background currents, volume and heat transport, and their evolution along the path of WSC. Vertical mixing and heat loss from the WSC are quantified. The

goal of this study is to improve the general understanding of processes modifying the Atlantic Water inflow, into the Arctic Ocean, and describe the importance of vertical mixing versus horizontal processes during summer. We propose that convective mixing in the bottom-boundary layer and the subsequent lateral export of mixed water can make a substantial contribution to the cooling rate of the WSC.

## 2  Data

The data set analyzed in this study was collected from the research vessel (RV) Håkon Mosby, between 12 and 21 August 2015. The ship track and the locations of the different stations are shown in Fig. 1b. Data were collected mainly along three sections, referred to as Section A, B and C, using the conductivity-temperature-depth (CTD) and lowered acoustic Doppler



current profiler (LADCP) system, the vertical microstructure profiler (VMP) and the shipboard acoustic Doppler current profiler (SADCP). In total, 46 CTD/LADCP and 85 VMP profiles are analyzed.

## 2.1 Temperature and salinity measurements

The CTD profiles were acquired using a Sea-Bird Scientific, SBE 911plus system. A 200-kHz Benthos altimeter allowed
profiles to within 10 m of the sea bed. The CTD system was also equipped with a WET Labs C-Star transmissometer. Accuracy of the pressure, temperature, and salinity sensors are $\pm0.5$ dbar, $\pm2 \times 10^{-3\,°}$C, and $\pm3 \times 10^{-3}$, respectively. The CTD data are processed using the SBE software following the recommended procedures. Conservative Temperature, $\Theta$ and Absolute Salinity, $S_A$ are calculated using the thermodynamic equation of seawater (IOC et al., 2010), and the Gibbs SeaWater (GSW) Oceanographic Toolbox (McDougall and Barker, 2011).

## 2.2 Current measurements

Horizontal current profile measurements were made using the LADCP and SADCP systems. All current measurements are corrected for the magnetic declination. Two 300-kHz Teledyne RD Instruments Workhorse LADCPs were installed on the CTD rosette collecting 1-s profiles in master–slave mode to ensure synchronization. The sampling vertical bin size was set to 8 m for each ADCP. The LADCP data are processed as 8-m vertical averages using both ADCPs and both up and down casts,
and using the Lamont-Doherty Earth Observatory (LDEO) Software version IX.12, which is an implementation of the velocity inversion method described in Visbeck (2002). Profiles are obtained using the constraints from velocities from ship navigation, bottom tracking and SADCP, with a resulting horizontal velocity error less than $3 \, \mathrm{cm\,s^{-1}}$ (Thurnherr, 2010).

     SADCP on RV Håkon Mosby was a 75-kHz Teledyne RD Instruments Ocean Surveyor. It collected velocity profiles continuously in the broadband mode. Final profiles, 5-minute time averaged, are obtained using the University of Hawaii software
(Firing et al., 1995). Typical final processed horizontal velocity uncertainty is 2–3 $\mathrm{cm\,s^{-1}}$.

## 2.3 Microstructure measurements

Ocean microstructure measurements were made using a 2000-m rated VMP manufactured by Rockland Scientific, Canada (RSI). The VMP is a loosely-tethered profiler with a nominal sink velocity of 0.6 $\mathrm{m\,s^{-1}}$. The profiler was equipped with pumped SBE-CT sensors, a pressure sensor, microstructure velocity shear probes, one high-resolution temperature sensor, one
high-resolution microconductivity sensor and three accelerometers.

     The processing of the microstructure data is based on the routines provided by RSI (ODAS v4.01) (Douglas and Lueck, 2015). Assuming isotropic turbulence, the dissipation rate of turbulent kinetic energy per unit mass can be expressed as

$$\epsilon = \frac{15}{2}\nu\overline{\left(\frac{\partial u}{\partial z}\right)^2} \tag{1}$$

where $\nu$ is the kinematic viscosity, overbar denotes averaging, and the $\partial u/\partial z$ is the small scale shear of one horizontal velocity
component $u$. Dissipation rates are calculated from the shear variance obtained by integrating the shear wavenumber spectra,



using 1-s FFT length and half-overlapping 4-s segments, following the corrections and methods described in the RSI Technical Notes (https://rocklandscientific.com /support/knowledge-base/technical-notes/). Resulting values are quality screened by inspecting the instrument accelerometer records and individual spectra from the two shear probes. Estimates from both probes are averaged when they agreed to within a factor of 10. Otherwise, the lower dissipation value is accepted, because larger

values can be caused by spikes induced, e.g., by impact with planktons.

## 3 Methods

### 3.1 Water masses

We use the classical categorization of water masses in the region, as first defined by Swift and Aagaard (1981), and later modified by Aagaard et al. (1985), listed in Table 1. Note, however, changes in the properties and distribution of the intermediate

and deep waters in Fram Strait were observed, and discussed in Langehaug and Falck (2012). The Absolute Salinity, $S_A$, in Table 1 is calculated from the practical salinity values at 80°N and 10°E, and rounded to the nearest hundredth.

### 3.2 Tidal currents and geostrophic currents

The barotropic tidal current components in the LADCP and SADCP profiles are removed using the 5-km horizontal resolution Arctic Ocean Tidal Inverse Model, AOTIM-5 (Padman and Erofeeva, 2004). The tidal transport at specified latitude and longi-

tude coordinates are predicted for the mid-time of the current profiles, and the barotropic tidal current is obtained by dividing by the water depth at that specific location. At the CTD/LADCP stations (Fig. 1), where station depth is accurately measured, the measured station depth is used. The water depth elsewhere (for SADCP) is obtained from the International Bathymetric Chart of the Arctic Ocean (IBCAO) database (Jakobsson et al., 2012).

Hydrography and current profiles collected along the three sections are gridded to 2-m vertical and 1-km horizontal distance,

using linear interpolation. After uniformly gridding the data, a moving average smoothing is performed using a 10 km × 10 m (horizontal × vertical) window. The smoothing is intended to remove the ageostrophic and short time and length scale variability.

Dynamic height anomaly and the geostrophic currents are calculated relative to a reference pressure of 100 dbar, from the gridded and smoothed $S_A$ and $\Theta$ fields. The reference pressure is chosen so that it is away from frictional boundary layers where ageostrophic currents can be substantial. The absolute geostrophic velocity is then obtained by adding the across-

section component of the observed currents at the reference level. The observed currents used are the detided LADCP profiles, identically gridded and smoothed as the hydrographic fields for consistency.

### 3.3 Stream tubes

The stream tube of WSC is defined using the absolute geostrophic velocities in the AW layer. The vertical extent of the tube is

defined by the AW layer (or sea bed). The center of the stream tube is defined as the location of the maximum layer-integrated





velocity (i.e. transport density, $\text{m}^2\,\text{s}^{-1}$), and assigned $x = 0$ km (see Fig. 6 and 5). The lateral extent of the stream tube is defined in two alternative ways.

In the first alternative (stream tube 1), the horizontal bounds are identified on either side of the core, as the location where the transport density first drops below a background threshold (solid enclosed curves in Figs. 2, 3, and 4). The threshold for

the transport density at each horizontal grid point is calculated by multiplying the AW layer thickness with $0.04\,\text{m s}^{-1}$ (i.e., a current estimate above the ADCP measurement error of $0.03\,\text{m s}^{-1}$). For sensitivity, $0.02\,\text{m s}^{-1}$ and $0.08\,\text{m s}^{-1}$ are also tested. The outer stations collected by the LADCP at Section C are separated by approximately 20 km, and the linear interpolation results in currents that deviate substantially from the SADCP observations (Fig. 5c). The depth-averaged currents from the SADCP better resolve the lateral structure, off slope away from the core, hence for the stream tube in Section C we use an

outer bound ( $x = -11$ km) obtained by applying the threshold to the SADCP velocity.

In the second alternative (stream tube 2), we conserve the volume flux (of the Svalbard branch) in the tube to within 10% at all sections (dashed enclosed curves in Figs. 2, 3, and 4). The volume flux of 1.3 Sv at Section A is deemed representative of the Svalbard branch (see Sect. 4.2), and used as a constraint in Sections B and C. The lateral edges of the tube are identified by integrating the volume flux at equal horizontal distance increments centered at the core (Section B is limited by shelf).

## 15 3.4 Heat Change

Neglecting molecular diffusion, the rate of change of heat content, $q$, of a body of fluid is balanced by the mean advection of heat and the eddy heat flux divergence

$$\frac{\partial \bar{q}}{\partial t} + \bar{\mathbf{u}} \cdot \nabla \bar{q} + \nabla \cdot \overline{\mathbf{u}'q'} = 0, \tag{2}$$

where $\mathbf{u} = \bar{\mathbf{u}} + \mathbf{u}'$ is the horizontal velocity, an overbar denotes averaging over several eddy time scales, and primes denote

fluctuations. Following the method described by Boyd and D'Asaro (1994) and Cokelet et al. (2008), we integrate Eq. 2 over a fixed volume, $V$, to obtain

$$\frac{\partial}{\partial t} \int_V \bar{q}\,dxdydz = \int_{Surface} (Q + H)\,dxdy - \int_V \bar{\mathbf{u}} \cdot \nabla \bar{q}\,dxdydz, \tag{3}$$

where eddy fluxes are neglected, and $Q$ and $H$ are the heat flux to the atmosphere and sea ice, respectively. Assuming that the local heat content does not change, the surface heat flux must balance the divergence of heat. Applying the Gauss theorem on

the volume integral of the heat advection, yields

$$\int_{Surface} (Q + H)\,dxdy = \rho_0 C_P \bar{v} \int_A \frac{\partial \overline{T}}{\partial y}\,dxdz, \tag{4}$$

where the area integral is taken over the current's cross section $A$, $\overline{T}$ is the mean temperature, $\overline{v}$ is the mean velocity normal to the section, $\rho_0$ is seawater density and $C_P$ is the specific heat capacity. Here, $y$ is the along-path coordinate, and estimated as





the distance along the 500-m isobath along West Spitsbergen from Section C to A (approximately 0, 86 and 171 km at Sections C, B, and A). We use $\Theta$ in the calculations, and the average temperature for each section is calculated as the velocity-weighted average over the stream tube. From Eq. 4, the surface heat loss per along-path meter (W m$^{-1}$) from the WSC to the layer above can be estimated. The along-path temperature gradient is obtained from the slope of a line fit to three data points: along-path

5   distance against the velocity-weighted average $\Theta$ for each section. The surface heat flux (W m$^{-2}$) is then obtained by dividing the heat loss by the width of the stream tube.

### 3.5 Turbulent heat fluxes

The turbulent heat flux, $F_H = \overline{w'T'}$, where $w'$ and $T'$ are the vertical velocity and temperature fluctuations, can be obtained as down-gradient diffusive flux:

$$F_H = -K_T \frac{\partial \bar{T}}{\partial z}, \tag{5}$$

where $\partial \bar{T}/\partial z$ is the mean vertical temperature gradient, $K_T$ is the eddy diffusivity for heat, and the averaging and fluctuations apply to turbulent eddy scales (different than in Eq.2). Assuming that heat and density diffuse with similar coefficients in a turbulent flow ($K_T \approx K_\rho$), the diapycnal eddy diffusivity can be obtained from the shear probe data using the Osborn (1980) relation

$$K_\rho = \Gamma \frac{\varepsilon}{N^2}, \tag{6}$$

where $\Gamma$ is the efficiency coefficient, $\varepsilon$ is the turbulent dissipation rate, and $N$ is the buoyancy frequency. The efficiency coefficient is variable and uncertain, but is commonly set to 0.2, which is the recommended value for typical oceanographic applications (Gregg et al., 2018). The heat flux is calculated from the shear measurements using $K_\rho$ with $\Gamma = 0.2$ and a vertical scale of 10 m for the background temperature and density gradients.

20  ## 4   Results and discussion

### 4.1   Hydrography

Conservative Temperature and Absolute Salinity distributions in Sections A, B and C, show the changes along the path of the WSC (Fig.2). The stream tubes defined in Sect. 3 are outlined in the hydrographic sections. Note that Section A is the northernmost and C is the southernmost section, and the horizontal distance is referenced to the location of the WSC core.

25      Temperatures near surface exceed 6°C in August, and decrease with depth in all three sections (Fig. 2). The northern part of Section A is close to the ice edge, and is characterized by cold surface waters. Compared to past observations, the August 2015 conditions were particularly warm, not only near the surface but also in the water column. In October/November 2001 the AW

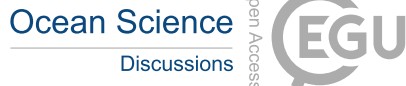



temperatures above 4°C, west of Svalbard, were separated from the surface by colder water (Cokelet et al., 2008), similar to the observations in winter 1989 (Boyd and D'Asaro, 1994), and in September 2012 north of Svalbard (81°30′N, 30°E) (Våge et al., 2016). When compared with the Monthly Isopycnal and Mixed-layer Ocean Climatology (MIMOC) (Schmidtko et al., 2013) in August (not shown), AW temperatures in 2015 were up to 1.8°C warmer. Similarly, observations made from drifting

pack ice north of Svalbard in spring 2015 showed warmer and shallower AW compared to the climatology (Meyer et al., 2016). A subsurface salinity maximum between 100 m and 400 m depth was found in all three sections, similar to previous studies in this region (Cokelet et al., 2008; Våge et al., 2016; Meyer et al., 2016). Compared to the climatology, the salinities were higher in all sections.

Water mass categorization (see Table 1 for definitions) shows that the AW layer overlaying LAIW extends across all sections

(Fig. 3). A temperature-salinity diagram analysis following Cokelet et al. (2008) (not shown) indicates that the formation of LAIW, UAIW and DW is dominated by atmospheric cooling, i.e., the ratio of heat loss to atmosphere and to sea ice ($Q/H$) exceeds 5, consistent with previous observations in winter (Boyd and D'Asaro, 1994) and autumn (Cokelet et al., 2008). In Section C, ASW is transformed by sea ice melting by AW ($Q/H = 0$). In the sections farther downstream (B and A), the ASW transformation process is complex, affected by mixing of warm and relatively fresh surface water from earlier melting events

with the AW in the upper water column.

An objective analysis of temperature and salinity at 100 dbar pressure, indicates that the AW properties extend along the 1000-m isobath northward. The same pattern is seen for the 100-600 m depth-averaged AW properties. However, the AW loses its depth-averaged temperature much faster than its salinity, implying mixing with colder water with similar salinity, such as LAIW.

## 4.2  Currents and transport

The spatial distribution of the currents measured by the SADCP helps identify the typical circulation patterns in the study region. Objectively interpolated depth-averaged currents from the SADCP show a well-defined Svalbard branch of the WSC along the 400-m isobath (Fig. 1b). The Yermak branch however, is not well-captured by the SADCP. Over the YP, there is no clear evidence of the Yermak Pass branch in this snapshot of observations. The absolute geostrophic currents toward the end of

Section B (west of $x = -75$ km) , however, show currents toward the Arctic (positive values), which can be a signature of the Yermak Pass branch. This branch is expected to be variable and weak in summer (Koenig et al., 2017). Between the Svalbard branch and (possibly) the Yermak Pass branch, a barotropic current is directed southwest, centered at $x = -65$ km (Fig. 4). North of the Molloy Hole, the currents are north-northwestward, consistent with the main recirculation route for the warmest AW (Hattermann et al., 2016).

The vertical distribution of the observed geostrophic currents along the sections captures the core of the WSC and its lateral extent (Fig. 4). The absolute geostrophic current profiles show a strong barotropic component. Sections A and C across the continental slope have a well-defined WSC core with a maximum velocity exceeding 0.3 m s$^{-1}$, typically located above the 400-600 m isobath. Section B, on the other hand, extends over the YP.



Vertically-averaged currents, between 50 and 500 m for consistency, are compared between LADCP and SADCP, as well as the AOTIM5 tidal currents (Fig. 5). While the core of WSC is densely sampled by the LADCP, the coverage in the outer parts of Section C is coarse. SADCP supplements the sampling. Note the segment between $x = -10$ and $-30$ km where the rapid lateral decay of the average current is not captured by the LADCP. In Section C, therefore, the stream tube 1 outer limit is

identified using the SADCP (Sect. 3.3). The agreement and coverage in other sections is good. While the currents in Sections A and C are substantially more energetic than tides, Section B is characterized by strong tidal currents, particularly over the plateau between $x = -30$ and $-80$ km.

Vertically-integrated currents and averaged temperature and salinity show that while the salinity maximum is approximately co-located with the geostrophic velocity maximum (the core), the temperature maximum is located landward (Fig. 6). The

depth-integrated velocities (volume transport per unit width) show a substantial peak at the core location (by definition), and an approximately symmetric lateral profile (Fig. 6). The lateral structure of the transport appears to be related to the location of the core relative to the slope. In sections A and C, the core is located over relatively even slopes, whereas in Section B the steep continental slope abruptly ends on the shallow YP, the current is less constrained with topography, and the volume transport calculations are sensitive to the stream tube boundary definition (Table 2).

When the volume transport is not constrained between sections, the transport in stream tube 1 is 2.8 Sv at Section C, comparable to the previous observations. Beszczynska-Möller et al. (2012) estimated that the long-term mean net volume transport of AW between 1997 and 2010, along the array of moorings at $78°50'$N, was $3.0 \pm 0.2$ Sv. The mean transport in August, averaged over 13 years, was $2.5 \pm 0.7$ Sv, with individual August averages as high as 3.6 Sv (Beszczynska-Möller et al., 2012). It should be noted that the volume transport estimated by Beszczynska-Möller et al. (2012) is for AW roughly

between the 2600-m isobath and the 300-m isobath. In contrast, the tube in Section C is located between the 1100-m isobath and the 140-m isobath. Furthermore, our observation is synoptic, and day-to-day or week-to-week variations are unknown.

Progressing along path, the volume transport and velocity-weighted average temperature in stream tube 1 are 0.7 Sv and 4.18°C in Section B, and 1.3 Sv and 3.64°C in Section A. Approximately 340 km downstream of Section A, at $81°50'$N and 30°E, Våge et al. (2016) estimated an AW volume transport of $1.6 \pm 0.3$ Sv in September 2012. This is larger than the

transport in Section A, but the error estimates overlap. Instead of salinity, Våge et al. (2016) used a density criterion to define AW ($27.70 \leq \sigma_\theta \leq 27.97$ kg m$^{-3}$ and $\Theta > 2°$C), following Rudels et al. (2005). As seen in Fig. 4a, these density bounds fit the stream tube well. Våge et al. (2016) included all AW with above-zero geostrophic velocity, whereas in stream tube 1 the boundary is drawn at 0.04 m s$^{-1}$; calculation using a threshold at 0.02 m s$^{-1}$, however, does not lead to an increase in the total transport. The barotropic currents in Section A are landward of the 800-m isobath, whereas Våge et al. (2016) found barotropic

currents 20 km seaward of the 800-m isobath. Overall, a small contribution from the Yermak Branch, flowing around the tip of the plateau and joining the slope boundary current, could contribute to an increase in volume transport and explain the off-slope extent of AW in Våge et al. (2016)'s observations.

If the inner branch of the WSC follows the $f/H$ contours, the partitioning of the Yermak and Svalbard branches in Section C can be estimated. The outer edge of stream tube 1 is located approximately at the 500-m isobath in Section B. The change

in Coriolis parameter between Sections A to C is negligible. The volume transport landward of the same isobath in Section





C (tube 1) is 0.6 Sv, which is within the uncertainty of volume transport in Section B. If the stream tube 1 in Section B is representative of the topographically-guided Svalbard branch of WSC, 0.6-0.7 Sv must flow through Section B toward A. Thus, the remaining 0.6-0.7 Sv could be delivered by the Yermak branch or the Yermak Pass branch in order to conserve volume. The volume transport landward of the 500-m isobath in tube 1 at Section A is 0.4 Sv, lower than Sections B and C.

To obtain 0.6 Sv in Section A, we have to extend the integration to a 100-m deeper isobath. Whether an event has caused the current to shift seaward, or whether the divergence in current along the continental shelf break (Fig. 1-Section A) has caused a decrease in volume transport is unclear. The lack of a well-defined slope on the shelf in Section B may lead to the break of topographic control and meander the current to deeper isobaths.

In the alternative definition of stream tube 2, motivated by the well-defined core structure of WSC at Section A, we assume
that the transport in Section A (defined by tube 1) is entirely the Svalbard branch, and this volume is conserved upstream at Sections B and C (i.e., stream tubes 1 and 2 are identical in Section A, and the volume transport is approximately 1.3 Sv in Sections B and C). This implies that 2.8-1.3 = 1.5 Sv of AW must be routed to the Yermak Branch after Section C. These figures are consistent with earlier observations and our present understanding of the AW circulation northwest of Svalbard.

### 4.3 Cooling and freshening of the WSC

The along-path cooling and freshening of the WSC inferred from the change of section-averaged (velocity weighted) properties from Section C to A are similar to previous observations in summer and fall. The change of salinity is -0.015 g kg$^{-1}$/100 km, comparable to the downstream freshening of 0.013/100 km (in practical salinity scale) reported by Cokelet et al. (2008), and the 50-year mean summer freshening of 0.010/100 km, measured obtained by Saloranta and Haugan (2004). The northward temperature gradient is -0.20 °C/100 km for stream tube 1, and -0.23 °C/100 km for stream tube 2. Cokelet et al. (2008)
observed -0.19°C/100 km in fall 2001. Saloranta and Haugan (2004) observed a 50-year summer mean cooling of 0.20°C/100 km, same as that observed in 1910, between 75°N and 79°N (Helland-Hansen and Nansen, 1912).

Calculation of the northward heat change using equation 4 requires conservation of volume, which is satisfied for stream tube 2, but not in tube 1. In calculations for the tube 1, we use the transport averaged over 3 sections. The bounds on estimates are obtained from calculations using tube widths from the 0.02 and 0.08 m s$^{-1}$ background velocity thresholds (Sect.3.3), which
reflect on velocity weighted averages, cross-section areas, as well as along-path gradients. For tube 1 we obtain an along-path heat change of -1.3 [-1.4 − -1.1]×10$^7$ W m$^{-1}$. When conserving the volume flux at 1.3 Sv, for tube 2, the heat change is -1.2 ×10$^7$ W m$^{-1}$. Dividing by the average tube width yields an estimate of the surface heat flux, resulting in 490 [460 550] W m$^{-2}$ for tube 1 and 380 W m$^{-2}$ for tube 2 (all rounded to the nearest 10 W m$^{-2}$).

Boyd and D'Asaro (1994) stated that a winter heat loss of $2 \times 10^7$ W m$^{-1}$ (within a factor of two) was needed to cool
the warm core as much as observed, comparable to but larger than the cooling rates we observe in summer. For comparison Saloranta and Haugan (2004) estimated a summer heat loss of 330 W m$^{-2}$, and Cokelet et al. (2008) reported 310 W m$^{-2}$. Both studies used a mean velocity of 0.1 m s$^{-1}$ when estimating the heat flux. In our observations the mean velocity was 0.15 m s$^{-1}$; scaling the heat flux in tube 1 by a factor of 1.5 yields a result comparable to that of Saloranta and Haugan (2004) and Cokelet et al. (2008).





### 4.4 Turbulent heat fluxes

At the time of the cruise, AW was located below the warmer and fresher ASW, and above the colder LAIW. The temperature decreased with depth, hence all diffusive heat fluxes were negative (directed downward). Cooling by vertical heat flux is possible due to flux divergence where loss at the bottom of a layer exceeds input from top. Figures 7 and 8 show temporal

mean of microstructure measurements at Station R1 and R4 respectively. In Section A and C, where AW is above colder LAIW, the vertical heat flux is larger through the top of the AW layer than through the bottom, resulting in net heating (Fig. 7e and 8e). The measured average heating of AW (by vertical fluxes alone) from Section A and C is 2 and 1 W m$^{-2}$, respectively. In Section B, the AW layer reaches sea bed, and the negative heat flux at the top of the layer contributes to warming the AW.

The mean turbulent heat fluxes within the AW layer are between -1 and -8 W m$^{-2}$, with the largest heat flux of -39 W m$^{-2}$

observed at station R3 in Section B. Away from the slope and bottom boundary layer, average heat fluxes are close to zero within the AW layer. The small fluxes north and south of the YP are consistent with previous findings in the Arctic region (Sirevaag and Fer, 2009; Krishfield and Perovich, 2005). Elevated fluxes over the YP are consistent with the observations of Padman and Dillon (1991) and Fer et al. (2010). Dissipation of TKE is small within the AW interior, only exceeding $10^{-8}$ W kg$^{-1}$ in the surface and bottom boundary layer (Figs. 7 and 8c). Overall, the turbulent heat flux is too small to account for the

cooling rate of the WSC inferred from our observations.

### 4.5 Lateral mixing and convectively-driven bottom boundary layer mixing

If the AW layer is not cooled by vertical mixing, processes such as lateral mixing, shelf-basin exchange, and intrusions of cold shelf water could play a role. Farther downstream, on the East Siberian continental slope, Lenn et al. (2009) argue that lateral mixing with shelf water must be one of the major causes for the observed evolution of the AW boundary current.

A mean current flowing along the slope in the direction of Kelvin wave propagation induces a downslope Ekman transport that advects lighter waters under denser waters, driving diapycnal mixing and extracting potential vorticity (Benthuysen and Thomas, 2012). Note that in rotating stratified flows over a slope, symmetric instabilities can develop before the flow becomes convectively unstable (Allen and Newberger, 1998). Using detailed measurements of vertical profiles of turbulence and density through the bottom boundary layer over a sloping continental shelf, Moum et al. (2004) documented energetic convectively

driven mixing induced by downwelling Ekman transport of buoyant bottom fluid.

Here we propose that convective mixing of the unstable bottom boundary layer on the slope, driven by Ekman advection of density beneath the core of the WSC, followed by the detachment of the mixed fluid and its transfer into the ocean interior (Armi, 1978) can play an important role in the modification of the WSC properties. Vigorous turbulent convection, associated with the generation of localized plumes of rising light fluid, could suspend sediments, leading to intermediate nepheloid layers –

middepth layers of elevated suspended sediment concentration laterally advected into deep water from nearby slopes (McPhee-Shaw and Kunze, 2002). Our observations are supportive of this scenario in all three sections, and here we present details from Sections C (Fig. 9) and A (Fig. 10).



The WSC core over the slope flows in the direction of Kelvin wave propagation, and the stratification toward the shelf is characterized by less dense, relatively fresh near-bottom waters. A vertical buoyancy flux to drive convection can thus be delivered by downslope Ekman advection of buoyant water across the slope (e.g., Moum et al., 2004). Concentrated between the 400 and 600-m isobaths, the rate of dissipation is elevated near the bottom 100 m, by two orders of magnitude above the

interior levels (Figs. 9 and 10b, vertical columns). In this segment of the slope, the light transmissivity is largely reduced, and extends offshore across the section (Figs. 9b and 10b, background shading). The reduction of transmissivity can be interpreted as the increase in concentration of suspended matter in the water, likely in response to vigorous turbulence and convection. The sediment-laden waters are exported laterally and isobarically, and appear to cross isopycnals. The light transmissivity measurements are in situ, not calibrated, and must be interpreted with caution. Nevertheless, the pattern is consistent and

significant in all sections.

A vertical profile in each section, at approximately the 600-m isobath, shows the correspondence of the turbulent bottom layer with strong near-bottom shear (approximately 0.1 m/s over 100 m vertical distance), quasi-homogeneous bottom density and the low-transmissivity layer (Fig 8c-d, 9c-d). A close-up of the bottommost 100 m shows that the turbulent layer is characterized by nearly well-mixed density (Figs. 9f and 10f), and a particularly low-stratified bottom layer of approximately

40-60 m thickness with slightly colder but less saline water. On the West Spitsbergen slope, beneath the WSC core in the near-bottom layer, both shear-generated turbulence and convectively-driven turbulence must play a role. Because convection fuels energy directly into the vertical component, we speculate that convection can substantially contribute to sediment suspension to yield the low transmissivity signature we observe. Order of magnitude calculations guided by observations support this scenario.

For three adjacent stations over the slope, the vertically-averaged temperature, salinity, and density are calculated in the bottommost 30 m. Average density anomaly is obtained relative to the bottom 30 m vertical average. The results are comparable for all sections. The unstable density anomaly minima range between $1 \times 10^{-4}$ and $1 \times 10^{-3}$ kg m$^{-3}$. The lateral gradient of the bottom-average density is $-2 \times 10^{-6}$ to $-5 \times 10^{-6}$ kg m$^{-4}$, with temperature contributing a factor of 3 to 4.5 times more than salinity. For the observed range of density anomaly and a plausible range of vertical thickness between 10 and 80 m, Rayleigh

number varies between $10^{10}$ and $10^{13}$, above the critical value of about $O(10^3)$ (Turner, 1973).

The bulk stratification, $N^2$, and shear-squared, $S^2$, obtained from the slope of linear fits of density and velocity profiles against depth in the bottom 100 m are $6 \times 10^{-7}$ s$^{-2}$ and $10^{-6}$ s$^{-2}$, respectively, resulting in a bulk Richardson number, Ri $= N^2/S^2$ of 0.6. The average Ri calculated over 8-m gradients is smaller, $O(0.1)$. We therefore expect shear-generated turbulence production, in addition to convection. When both stress and the unstable buoyancy flux produce TKE, Lombardo

and Gregg (1989) show that the dissipation rate scales with a combination of the two sources. If roughly half of the dissipation is supplied by the convective buoyancy flux, the required cross-slope advection of buoyancy by bottom Ekman transport is $O(10^{-8})$ W kg$^{-1}$. For a range of Ekman layer thickness 1 to 5 m (corresponding to an eddy viscosity of $10^{-4}$ to $2 \times 10^{-3}$ m$^2$ s$^{-1}$), an average geostrophic current of 0.3 m s$^{-1}$ and the observed range of across-slope bottom-density gradient, the lateral buoyancy flux is in the range between 0.3 and $4 \times 10^{-8}$ W kg$^{-1}$, sufficient to maintain the observed dissipation rates.

The buoyancy flux of $O(10^{-8})$ W kg$^{-1}$ corresponds to a heat flux of approximately 400 W m$^{-2}$.





### 4.6 Isopycnal diffusion in an eddy field

In winter, an energetic eddy field diffuses heat along steeply sloping, outcropping isopycnal surfaces, at a rate sufficient to cool the subsurface warm core capped by stratification above (Boyd and D'Asaro, 1994). In winter, isopycnals in the core outcrop 5-10 km to the west of the core, whereas in our summer observations, the $\sigma_\theta = 27.7$ surface, an approximate upper bound of

the AW, is relatively flat in the outer part of the sections (Fig.2). Furthermore, the summer air temperatures are not expected to drive substantial heat loss to the atmosphere. Hence, we do not expect a large contribution from this process in summer. The lateral mixing by eddies, however, can still be substantial, depending on the dynamics of the eddy field and the structure of the isopycnals. The contribution from isopycnal diffusion as a result of barotropic instability could account for approximately 1/3 of the typical along-shelf cooling rate (Teigen et al., 2010). Baroclinic instability, most pronounced during winter/spring,

could lead to a heat loss from the WSC core, reaching 240 W m$^{-2}$ (Teigen et al., 2011). Both studies are highly idealized, but suggest that isopycnal diffusion by eddies can be important. Our data set is not sufficient to provide an accurate estimate of the isopycnal diffusion by eddies.

Våge et al. (2016) observed an anticyclonic eddy with a radius of 10-15 km and a vertical scale of 250-300 m. Crews et al. (2018) analyzed 177 eddies detected over the course of 2 years using eddy-resolving numerical experiments in the region north

of Svalbard. The eddies in the region can be characterized by a radius of 5-6 km, thickness of 300 m and average velocity anomaly of approximately 5 cm s$^{-1}$, overall consistent with the limited observations. Using an eddy length scale of $l' = 5$ km and velocity perturbation of $u' = 5$ cm s$^{-1}$, gives an estimate of the isopycnal diffusivity of $u'l' = 250$ m$^2$ s$^{-1}$. The average lateral temperature gradient of AW 0.05°C per km then yields an average horizontal flux of $5 \times 10^4$ W m$^{-2}$. Multiplying by a typical eddy thickness of 300 m, we obtain $(1-2) \times 10^7$ W m$^{-1}$, lost laterally outward along the side of the stream tube. This

is of the same order as the estimated along path heat change. Assuming a fraction (0.5) of the along path distance is affected by eddies, will result in half of the heat loss unexplained, unaccounted by interior diapycnal mixing and eddy-induced isopycnal diffusion. The cooling induced by convectively-driven bottom mixing can thus be important.

### 5 Conclusions

Observations from a cruise conducted in August 2015 provide a snapshot of the West Spitsbergen Current (WSC) hydrography,

transport and mixing, in ice-free regions west and north of Svalbard, in a year with relatively warmer and saltier Atlantic Water (AW) compared to climatology. Data were collected in three sections across the WSC, along an approximately 170 km path.

The Svalbard branch of the WSC is topographically guided close to the shelf break, and is relatively well captured by our observations. The recirculating branch in Fram Strait, the Yermak branch and the Yermak Pass branch are not resolved, but inferences are made from the observed sections. The volume transport in a stream tube of WSC, defined using the AW properties

and along-path velocities above a background value, is 2.8 Sv in Fram Strait before the Svalbard and Yermak branches split. The transport reduces to 1.3 Sv in the most downstream section north of Svalbard. In a scenario where the Svalbard branch is constrained to the 500-m isobath in the middle section, its volume transport is 0.6-0.7 Sv, implying approximately 2 Sv is fed to recirculation and Yermak branch in Fram Strait, and 0.6-0.7 Sv is potentially delivered by the Yermak branch or the Yermak





Pass branch which could join the Svalbard branch in the northern section. An alternative scenario, assuming the transport in the most downstream section is entirely the Svalbard branch (1.3 Sv) and this volume is conserved in the sections upstream, implies that 1.5 Sv of AW must be routed to the Yermak Branch in Fram Strait.

The along-path cooling of the WSC is similar to previous observations in summer and fall, with approximately -0.20 °C

per 100 km. The associated bulk heat loss per along-path meter is $(1.1\text{-}1.4)\times10^7$ W m$^{-1}$, corresponding to a surface heat flux of 380-550 W m$^{-2}$. The measured turbulent heat flux is too small to account for this cooling rate. In contrast to winter conditions, we do not expect heat loss by diffusion along outcropping isopycnals. The lateral mixing by eddies, however, can be substantial, depending on the dynamics of the eddy field and the structure of the isopycnals. While our observations are not sufficient to allow an accurate quantification of the diffusion by eddies, estimates using plausible range of parameters suggest

that the contribution from eddies could be limited to one half of the observed heat loss. We propose that energetic convective mixing of the unstable bottom boundary layer on the slope, driven by Ekman advection of density beneath the core of the WSC, followed by the detachment of the mixed fluid and its transfer into the ocean interior can explain the remaining heat loss.

The WSC core on the slope flows in the direction of Kelvin wave propagation, and the stratification toward the shelf is characterized by relatively less dense near-bottom waters. Our detailed observations show turbulence generation through a

combination of mean shear and convection. Downslope Ekman advection of buoyant water across the slope leads to a lateral buoyancy flux is in the range between 0.3 and $4\times10^{-8}$ W kg$^{-1}$, sufficient to maintain a large fraction of the observed dissipation rates. A buoyancy flux of $O(10^{-8})$ W kg$^{-1}$ corresponds to a heat flux of approximately 400 W m$^2$. Convectively-driven bottom mixing can lead to substantial cooling of the WSC, at a rate comparable to that expected from diffusion by eddies.

*Data availability.* The data set is available through the Norwegian Marine Data Centre. (DOI will be supplied upon acceptance of the paper.)

*Competing interests.* The authors have no competing interests.

*Acknowledgements.* This study was supported by the Research Council of Norway through the project 229786. The work is based on the M.Sc. study of EK (Kolås, 2017). We thank the crew and participants of the cruise HM 2015617.





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





**Table 1.** Water masses as defined by Swift and Aagaard (1981) and Aagaard et al. (1985). Absolute Salinity ($S_A$) is calculated from the practical salinity ($S$) at 80°N and 10°E, and rounded to the nearest hundredth.

| Abbr. | Name | Practical Salinity $S$ | Absolute Salinity $S_A$ $(g\ kg^{-1})$ | Conservative Temperature $\Theta$ (°C) |
|---|---|---|---|---|
| AW | Atlantic Water | $S > 34.88$ | $S_A > 35.05$ | $\Theta \geq 2$ |
| LAIW | Lower Arctic Intermediate Water | $S > 34.88$ | $S_A > 35.05$ | $2 > \Theta \geq 0$ |
| UAIW | Upper Arctic Intermediate Water | $34.88 \geq S \geq 34.7$ | $35.05 \geq S_A \geq 34.87$ | $\Theta < 1$ |
| DW | Deep Water | $34.96 \geq S > 34.88$ | $35.13 \geq S_A > 35.05$ | $\Theta < 0$ |
| PW | Polar Water | $S < 34.4$ | $S_A < 34.56$ | $\Theta < 0$ |
| PIW | Polar Intermediate Water | $34.7 > S \geq 34.4$ | $34.87 > S_A \geq 34.56$ | $\Theta < 0$ |
| ASW | Arctic Surface Water | $S < 37.4$ | $S_A < 34.87$ | $\Theta > 0$ |
| | | $34.88 \geq S \geq 34.7$ | $35.05 \geq S_A \geq 34.87$ | $\Theta > 2$ |



**Table 2.** Properties in stream tubes 1 and 2 in Section A (northernmost), B and C (southernmost). The bounds for stream tube 1 given in square brackets are calculated using tube widths from the background velocity thresholds of 0.02 m s$^{-1}$ and 0.08 m s$^{-1}$ (no bracket means no change in value). Stream tube 2 conserves a volume transport of 1.3 Sv to within 10%. Stream tube averages, indicated by overbars, are velocity-weighted.

| Section | A | A | B | B | C | C |
|---|---|---|---|---|---|---|
| Stream tube | 1 | 2 | 1 | 2 | 1 | 2 |
| $\Theta_{max}$ (°C) | 5.66 [5.85 5.57] | 5.66 | 6.38 | 6.38 | 7.44 | 7.45 |
| $\overline{\Theta}$ (°C) | 3.64 [3.62 3.67] | 3.64 | 4.18 [4.17 4.21] | 3.95 | 3.98 [3.96 3.99] | 4.03 |
| $S_{A max}$ (g kg$^{-1}$) | 35.24 | 35.24 | 35.26 | 35.26 | 35.28 | 35.28 |
| $\overline{S_A}$ (g kg$^{-1}$) | 35.20 | 35.20 | 35.22 | 35.14 | 35.23 | 35.24 |
| $\overline{v_g}$ (m s$^{-1}$) | 0.15 [0.13 0.19] | 0.15 | 0.10 [0.10 0.12] | 0.06 | 0.20 | 0.25 |
| **Area** (km$^2$) | 7.9 [9.4 5.8] | 7.9 | 7.3 [8.5 4.0] | 22.9 | 12.6 [13.7 11.5] | 5.5 |
| **Transport** (Sv) | 1.3 [1.3 1.1] | 1.29 | 0.7 [0.7 0.4] | 1.20 | 2.8 [3.0 2.6] | 1.36 |
| **Width** (km) | 24 [29 17] | 24 | 21 [24 11] | 61 | 35 [37 32] | 10 |



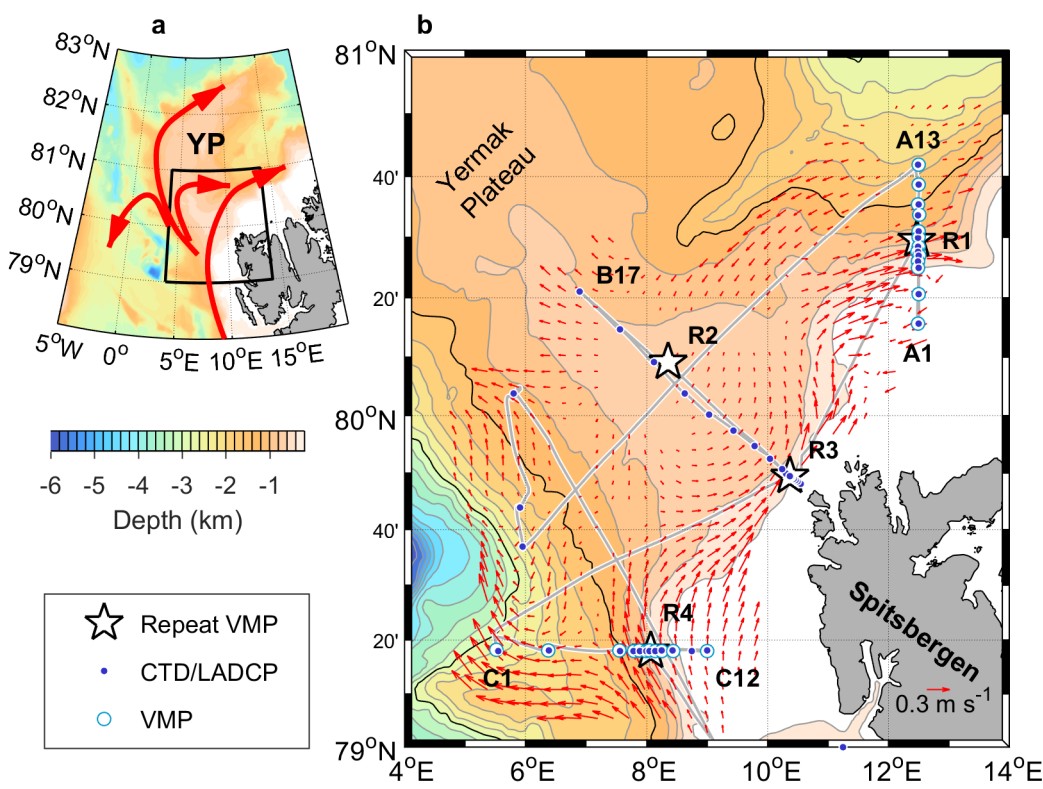

**Figure 1.** (a) Overview map with the AW circulation patterns (red arrows). The study region zoomed in panel (b) is indicated with a black rectangle. (b) Station locations of Sections A to C and the repeat stations R1 to R4. Selected station names at the edge of sections are marked for reference. Gray lines show the ship track during the cruise. Red arrows show the SADCP data averaged in the upper 500 m, objectively interpolated using a covariance function depending on the spatial distance between binned observations and their $f/H$ gradient following Böhme and Send (2005), using a 50-km correlation length scale and 5% error.



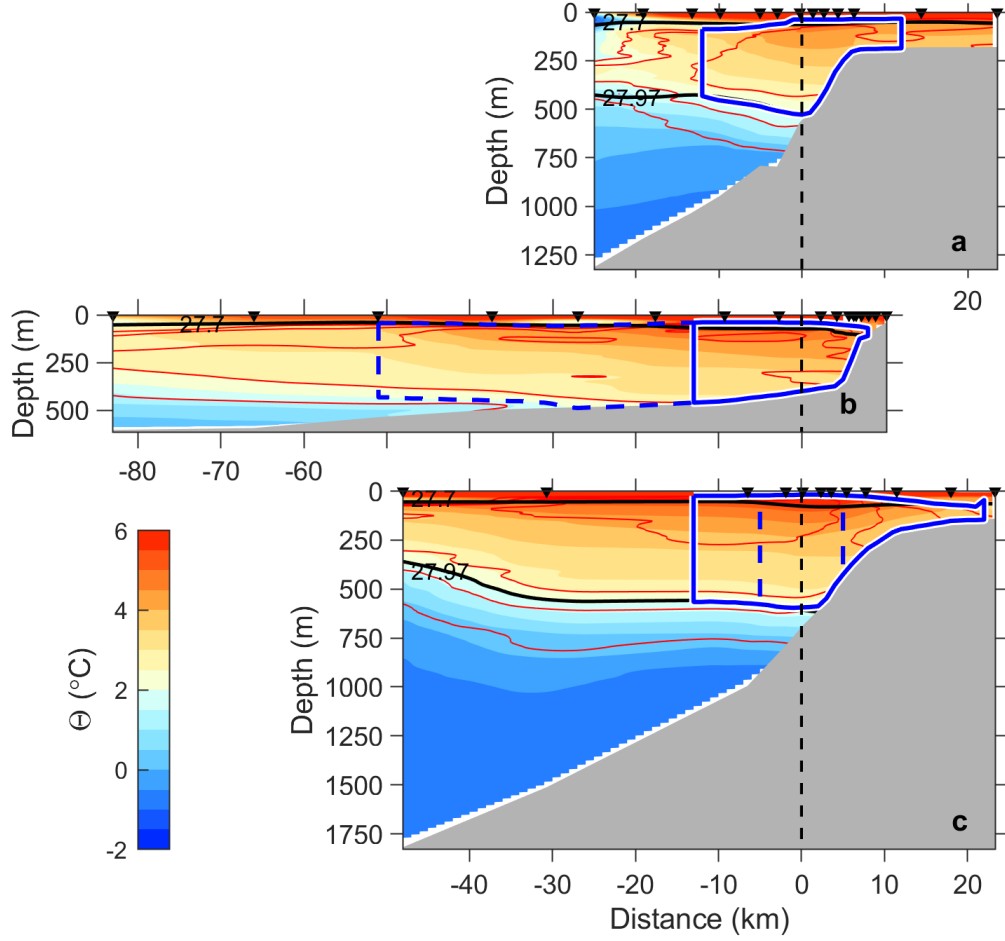

**Figure 2.** Distribution of Conservative Temperature ($\Theta$, color) and Absolute Salinity ($S_A$, red contours in 0.05 g kg$^{-1}$ intervals) along Section a) A, b) B and c) C. Blue lines enclose stream tube 1 (solid) and stream tube 2 (dashed). Black lines are potential density anomaly, $\sigma_\theta$, of 27.7 kg m$^{-3}$ and 27.97 kg m$^{-3}$. The horizontal distance is referenced to the core ($x = 0$) and the three sections are aligned at $x = 0$ (vertical dashed line). Station locations are marked with black arrows at the top of each panel.




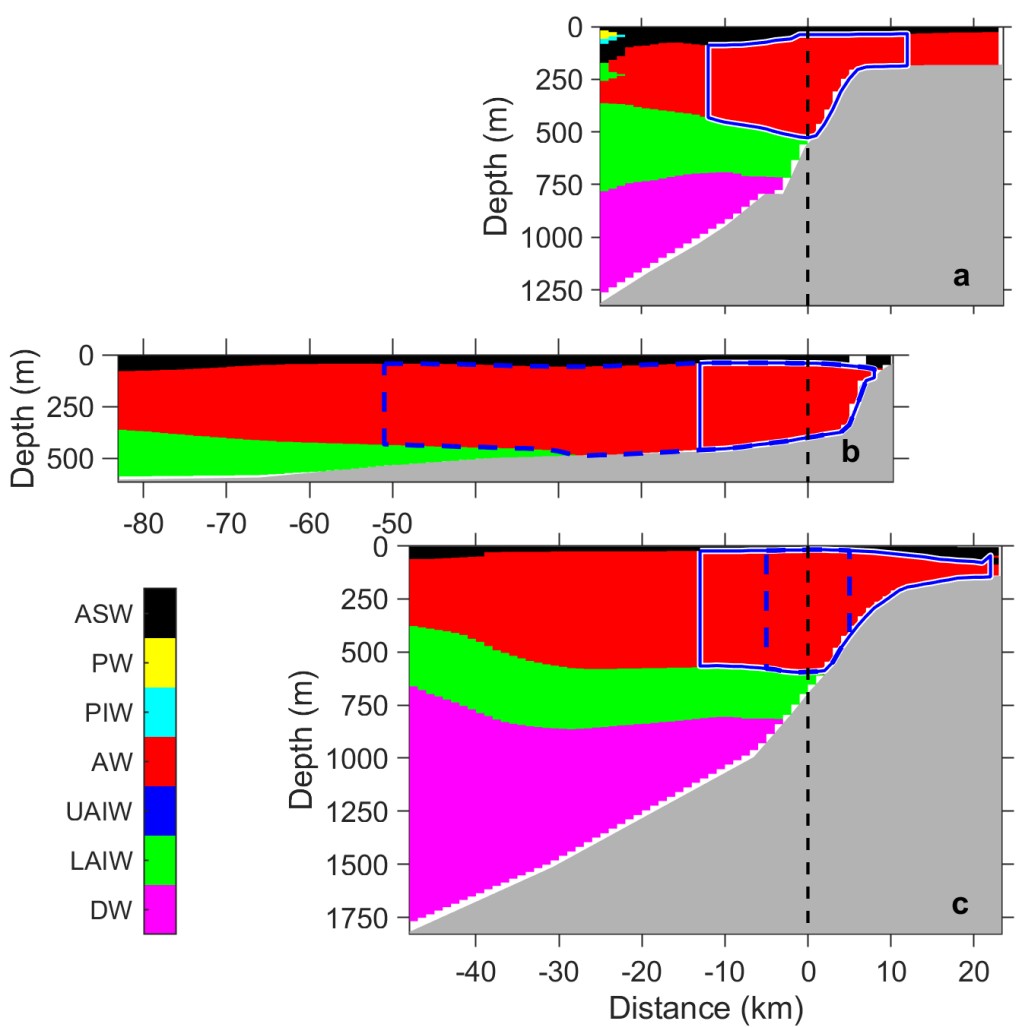

**Figure 3.** Distribution of water masses for Section a) A, b) B and c) C. Other details are as in Fig. 2. The different water masses are listed in Table 1.





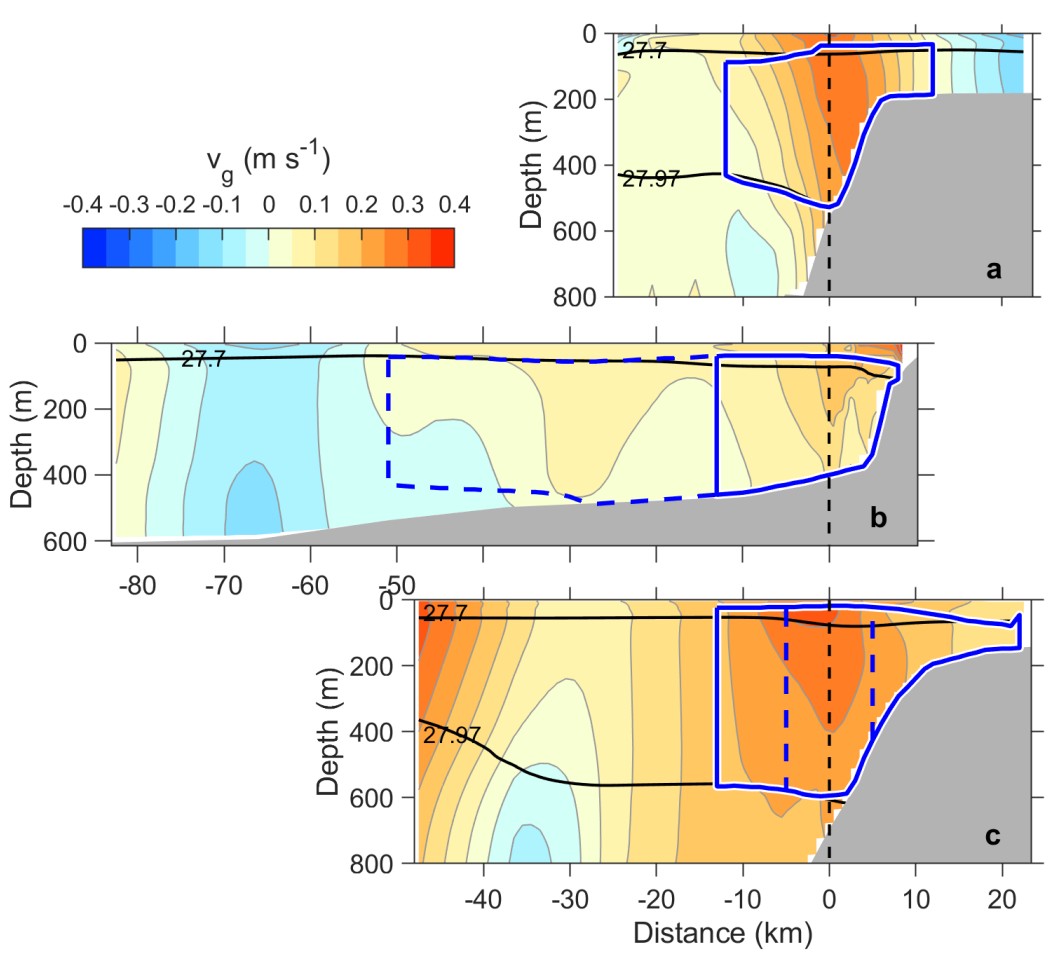

**Figure 4.** Distribution of absolute geostrophic velocity, $v_g$, along Section a) A, b) B, and c) C. Other details are as in Fig. 2.





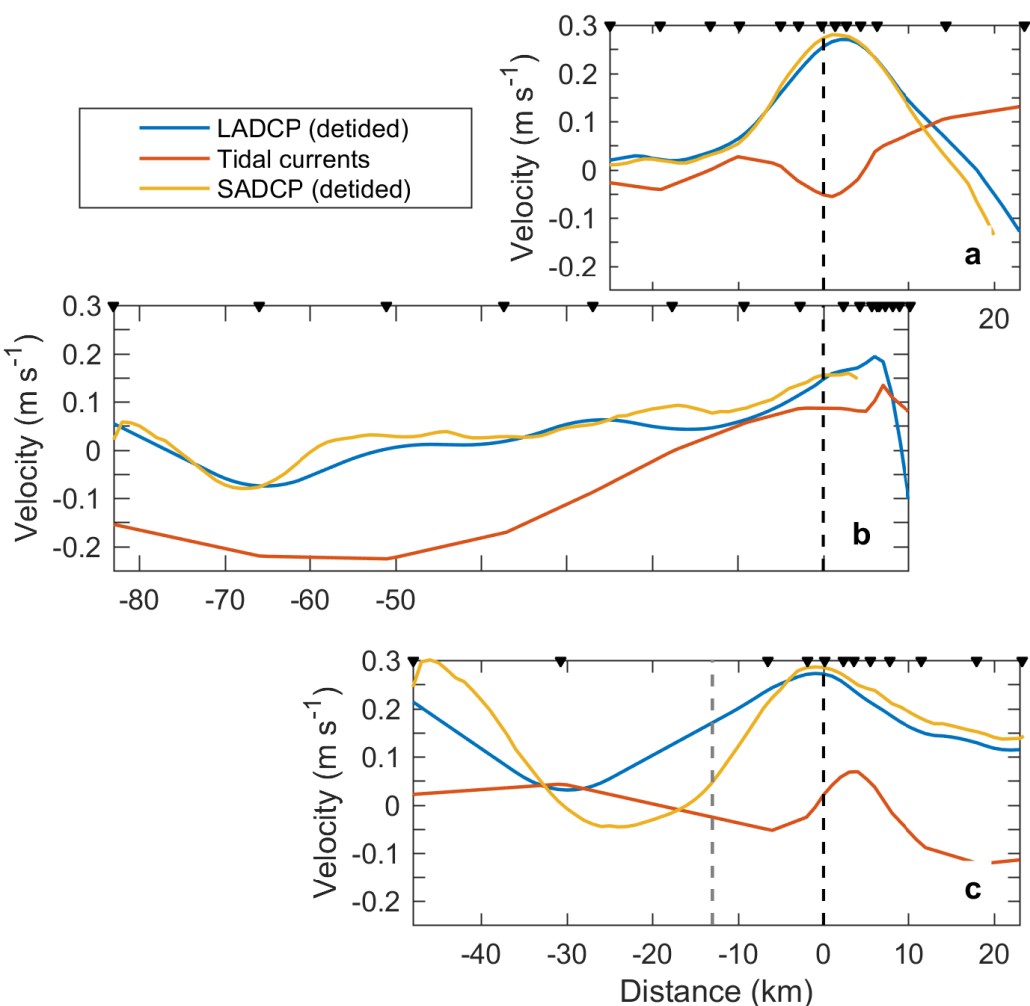

**Figure 5.** De-tided LADCP (blue) and SADCP velocities (yellow) vertically averaged between 50 and 500 m depth, and tidal currents (red) for Section A (a), B (b) and C (c). The dashed gray line at -11 km in (c) shows the outer bound of the stream tube 1 in Section C, based on SADCP measurements. Other details are as in Fig. 2.





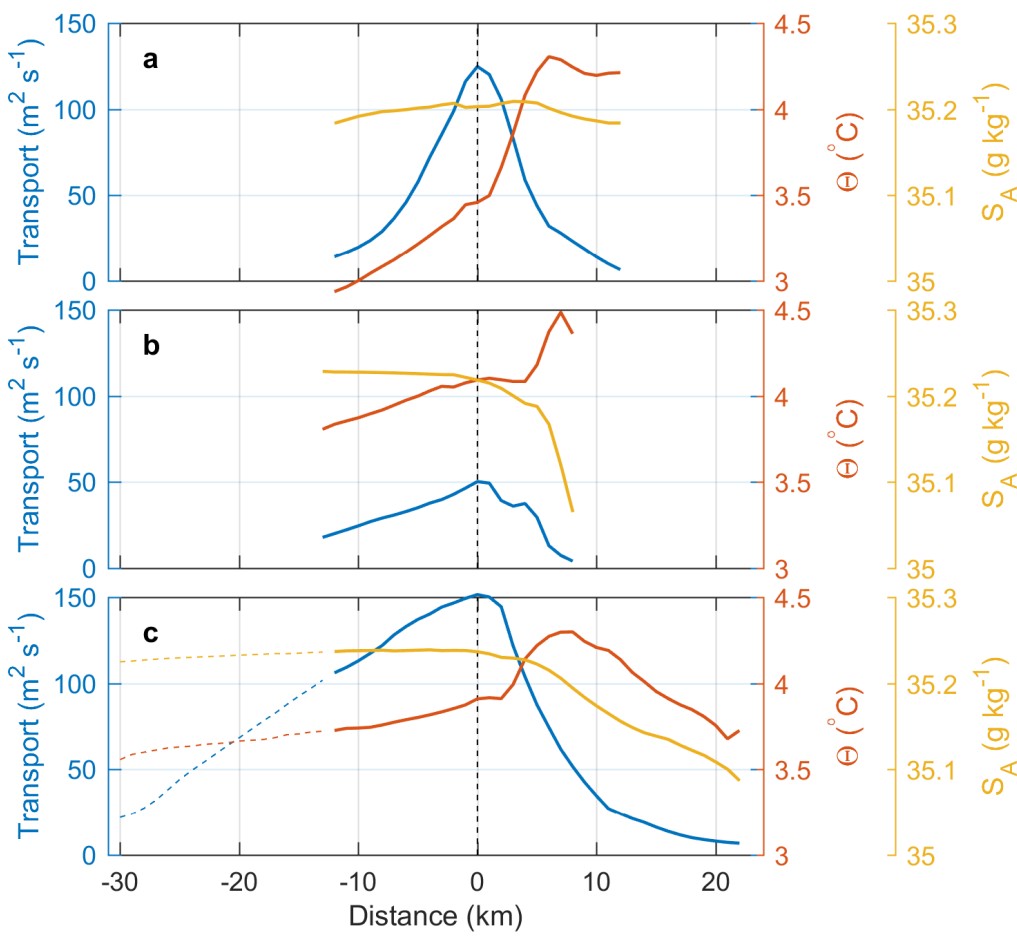

**Figure 6.** Vertically-integrated geostrophic velocity (transport per unit width, blue), and vertically-averaged $\Theta$ (red) and $S_A$ (yellow), for Section a) A, b) B, and c) C. The horizontal distance is referenced to the core at $x = 0$.





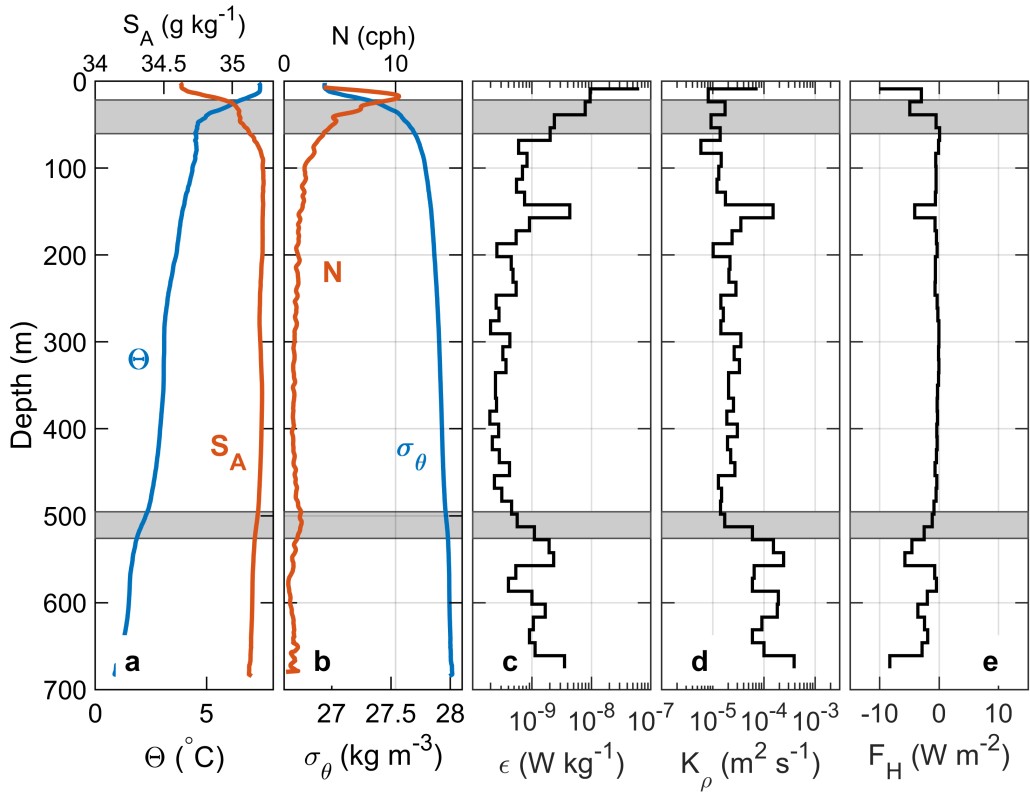

**Figure 7.** Temporal mean of measurements collected by the microstructure profiler at station R1, Section A. (a) $\Theta$ (blue) and $S_A$ (red), (b) $\sigma_\theta$ (blue) and buoyancy frequency $N$ (red), (c) 15-m bin-averaged dissipation of TKE $\epsilon$, (d) eddy diffusivity $K_\rho$, and (e) vertical heat flux $F_H$. Shaded bands envelop the upper and lower boundaries of the stream tube.



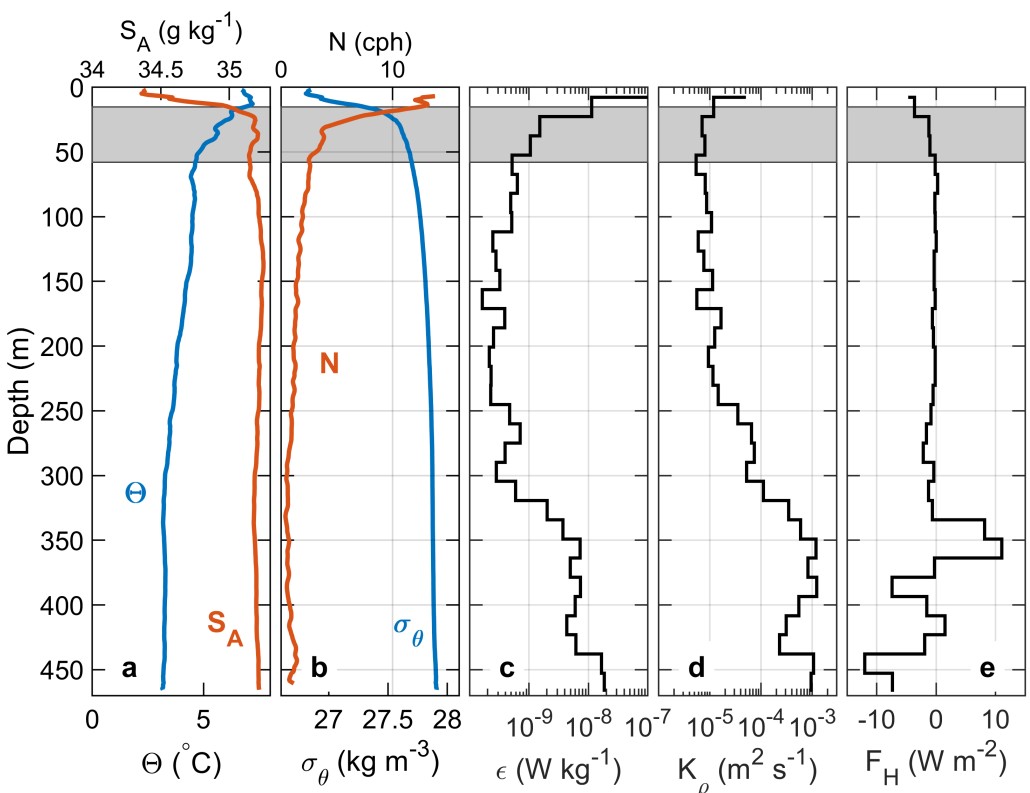

**Figure 8.** Same as Fig. 7, but for station R4, Section C.





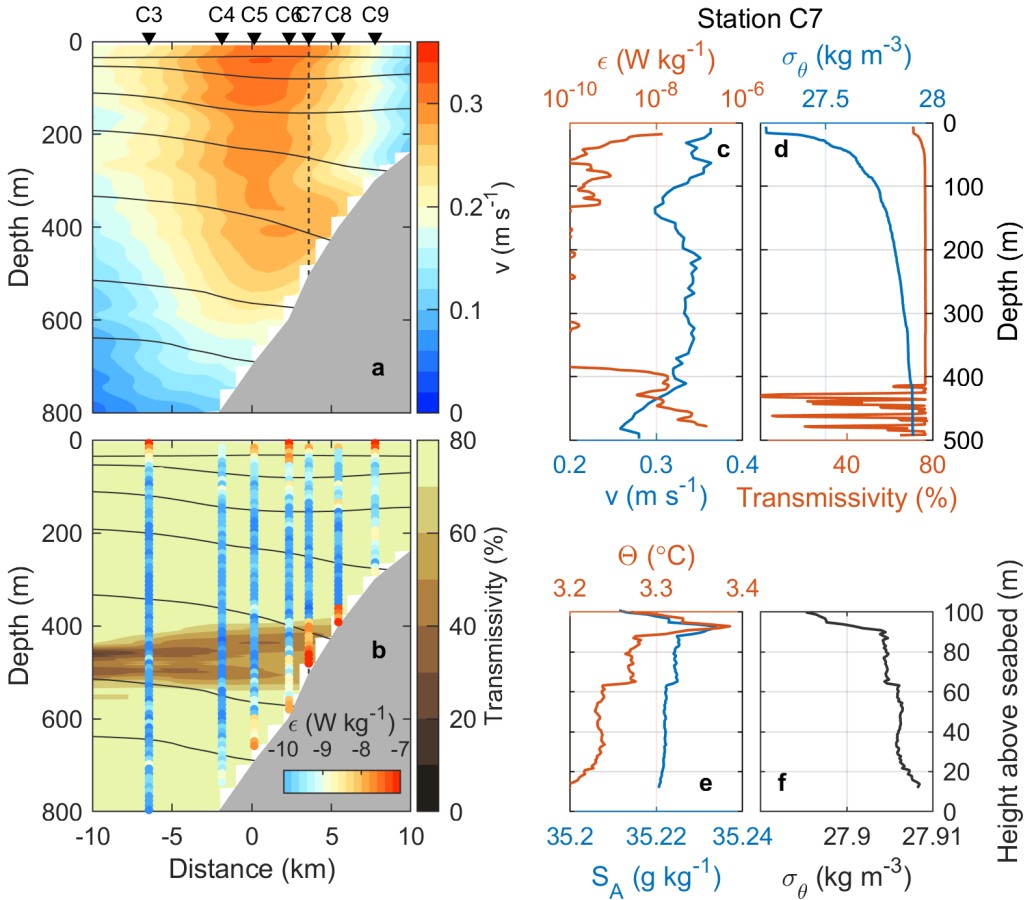

**Figure 9.** Distribution of a) the velocity component into the section and b) light transmissivity, together with selected isopycnals (black). Dissipation rate is color coded in vertical columns in (b). Profiles collected at station C7: full depth profiles of c) dissipation rate and velocity and d) $\sigma_\theta$ and transmissivity, and the bottom 100 m profiles of e) $\Theta$ and $S_A$ and f) $\sigma_\theta$.

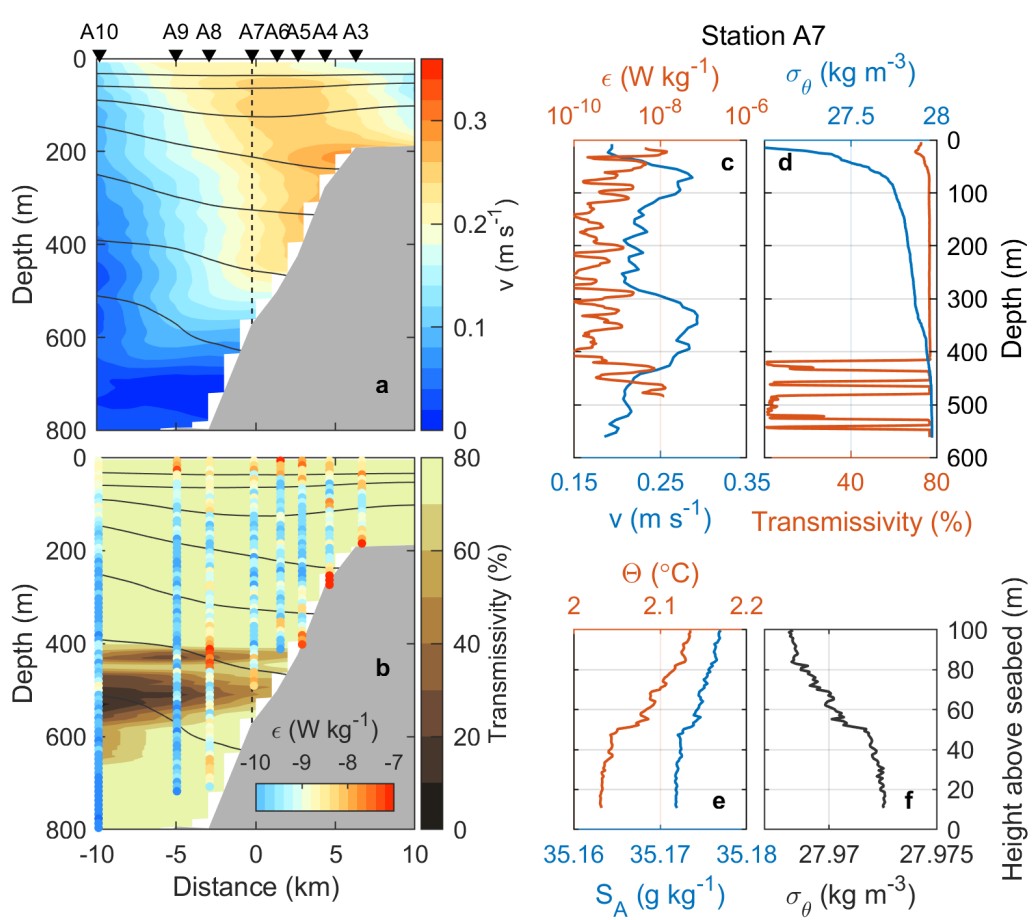

**Figure 10.** Same as Fig. 9, but for Section A and station A7.