# Peer review of "Hydrography, transport and mixing of the West Spitsbergen Current: the Svalbard Branch in summer 2015"

_Ocean Science, 2018_

## Referee Comment (RC1) · I. Polyakov (Referee) · 8 Oct 2018

R E V I E W

Journal: OS Title: Hydrography, transport and mixing of the West Spitsbergen Current: the Svalbard Branch in summer 2015 Author(s): Eivind Kolås and Ilker Fer MS No.: os-2018-86 MS Type: Research article

This is this rear occasion when reviewing a manuscript is fun work. The topic of the study is important and well thought. Data set is beautiful and in hands of skillful researchers. Analysis is thorough. Findings are solid and provide a new insight into

dynamics in the complex area under consideration. The manuscript is well and clearly written and presented materials are sufficient to illustrate conclusions made by the authors.

I congratulate the authors with such a nice piece of work.

I have just a few minor comments and suggestions and recommend publication after very minor changes.

Minor comments: 1. P. 3, lines 15-20. Any way to merge these conflicting estimates? 2. P. 5, Table 1: Can you please add mean vertical limits for the water masses? 3. P. 10. Line 29: Should the power be -2? 4. P. 11, line 21: "Extracting" is not a clear word here. 5. P. 11, lines 22-23: I do not see a point for placing this sentence ("Note that …") here. 6. P. 11, line 30: Should it be comma instead of "-" after McPhee? 7. P. 12, line 35. How was the correspondence between buoyancy and heat fluxes found? 8. Fig. 10b: Any comment on elevated dissipation rate at ~400m in the third deepest profile (i.e. in the interior) and lack of such a signature in other profiles?

---

## Referee Comment (RC2) · Anonymous Referee #2 · 16 Oct 2018

Review of "Hydrography, transport and mixing of the West Spitsbergen Current: the Svalbard Branch in summer 2015" by Kolas&Fer

The manuscript presents data from three shipboard sections occupied in summer 2015 northwest of Svalbard. The analysis focuses on explaining the along-pathway cooling of the West Spitsbergen Current concluding that vertical turbulent heat flux is not sufficient and that bottom mixing with shelf waters also makes a contribution. This material is relevant for a better understanding of the inflow of warm Atlantic Water to the Arctic Ocean and its modification processes along the inflow. This is appropriate and interesting for the readership of Ocean Science. The manuscript is well written and presents

the results in an easily digestible way even though the presented observations and analyses are complex. In my opinion, the authors missed one point in the discussion of their results which I would like the authors to comment upon/include in their discussion. Otherwise, the manuscript should be accepted after the correction of a few small suggestions. Therefore, I recommend minor revision.

Main comment: Inherent to the your interpretation of the presented results are the assumptions that the current is uniform in the along pathway direction, that there is no temporal variability in the current, and that your observations sampled the current at the same time. This does not hold for two reasons: There is strong mesoscale variability and individual shipboard sections may even capture substantial southward flow in the WSC (e.g. Richter et al 2018 Ocean Science). Therefore, one would not necessarily expect volume conservation between consecutive (in along pathway distance) synoptic sections. Put another way, differences between the transport in consecutive sections do not need to correspond to volume transport loss from the current. Furthermore, there is a seasonal cycle in the temperature of the WSC and its extension. The maximum in temperature of the Atlantic Water (well below the highly stratified low salinity cap on top of the current) is reached later in the season north(east) of Svalbard (recent A-TWAIN results) compared to the WSC west of Svalbard. In my opinion this is mainly due to the northward advection of a characteristic seasonal cycle set further south. For example, consider that the seasonal cycle has a slope of 2.5°C in 5 month (=0.5°C/month) west of Svalbard (e.g. von Appen et al 2016 Journal of Physical Oceanography). If one – for the sake of argument – considers a mean advective velocity of 0.1m/s (also quoted on page 10 line 32), the advection from your section C to your section A (distance of 170km) would take approximately 19 days. In those 19 days, the current (due to the seasonal cycle) at the southern location would have warmed by approximately 0.3°C. Put another way, at any one point in time (prior to the seasonal maximum in AW temperature), one would expect it to be 0.3°C cooler at section A than at section C. This corresponds to a horizontal temperature gradient of 0.18°C/100km, which is a number not substantially different from your 0.2°C/100km (page 10 line 19).

On top of that comes that your different sections were obviously not occupied on the same day, but with some (consider specifying explicitly) days in between them. I do not want to claim that this explains everything which you see and your interpretation is probably still broadly appropriate for what is happening to the current. Nevertheless, I think it would be worthwhile to discuss these points and to carefully consider where they might (and where they would not) impact your conclusions.

p1l11 Consider adding "We conclude that – at least in summer – convectively-driven bottom mixing. . ."

p1l12 Consider adding "can lead to substantial cooling and freshening of the WSC"

p2l21 Consider adding "These eddies may control"

p2l33 Note that these heat losses (in W/mˆ2) are dependent on the mean advective speed resp. the residence time of the water in the area of cooling.

p4l29 "averaging" Is this averaging in space or in time? Over what distance?

p5l21 Note that the smoothing does not necessarily remove all ageostrophic motions.

p5l28 Before this paragraph might be a good time to present the info on how long it took to occupy the individual sections. Is there e.g. contamination from multiple tidal cycles possibly being represented incorrectly?

p6l2 Here the considerations of my main comment come into play.

p6l26 Does this not require that the transport in the stream tube is exactly constant, not just to +-10%?

p7l27 e.g. Richter et al 2018 Ocean Science present data from 2016 which is similarly warm both near the surface and at greater depths as your observations.

p8l8 Consider "were higher in all sections by (range)." And then provide a range of salinity values.

p8l17-19 It might be helpful (though not necessary) to show a TS plot to better make this point.

p8l24 At what x-value in your plots would the YP branch supposedly be if it were present?

p9l33 Consider "If . . . follows the f/H contours and there is no synoptic variability between the sections, . . ."

p10l20 Be consistent with using "-0.20°C" versus "0.20°C" in this paragraph.

p11l3 "all vertical diffusive . . . (i.e. directed . . .)"

p11l6 "top", "bottom" It is not sure which values you refer to. Quote the numbers in your text.

p11l17 "If it is not cooled" Does it have to be cooled?

p12l9 Consider "the qualitative pattern"

p12l14 "weakly-stratified"

p12l35 Vertical or horizontal heat flux?

p13l2 "In winter . . ." Where does this supposedly (based on the reference) happen?

p13l18 Where/how was this average lateral temperature gradient estimated?

p14l14 "less dense waters" Where do these waters come from? How is that pool sustained? You could either elaborate on this a bit in the introduction or (more valuably) you could use this information to do some speculations on how changes in the sources of these water might affect the processes that you looked at/the cooling fo the WSC.

p15l29 This reference appears to be incomplete.

p18 LAIW line: The distinction between ">" and ">=" is meaningless for a continuous distribution.

Fig2/3 The near surface stratification due to salinity is really hard to see in this (Fig2) kind of figure. Consider plotting salinity instead of Fig3. The benefit of Fig3 is not obvious.

F6 Would this figure not be more effective with three subplots (1 for transport, 1 for temperature, 1 for salinity) and to have three differently colored lines in each of the subplots for each of the three sections.

Fig7/Fig8 Is 700m equal to the bottom depth here? Consider to have the same spacing (in centimeters on the paper) of the y axis in both Fig7 and Fig8.

Fig7 caption: "Shaded bands" Why are these bands so wide?

---

## Author Comment (AC1) · 18 Nov 2018

**Response to reviewers' comments on Kolås and Fer #os-2018-86, «Hydrography, transport and mixing of the West Spitsbergen Current: the Svalbard Branch in summer 2015".**

We thank both reviewers for their constructive comments and useful suggestions, which helped to improve the manuscript. Below, we provide a point-by-point response to all comments raised by the reviewers. Reviewers' comments are reproduced in italic type in red followed by our response in regular type, black color. We also now make the entire data set (CTD/LADCP/SADCP/VMP) openly available through

*Data availability.* The data set is available through the Norwegian Marine Data Centre, https://doi.org/10.21335/NMDC-567625440.

**Response to Reviewer 1:**

*This is this rear occasion when reviewing a manuscript is fun work. The topic of the study is important and well thought. Data set is beautiful and in hands of skillful researchers. Analysis is thorough. Findings are solid and provide a new insight into dynamics in the complex area under consideration. The manuscript is well and clearly written and presented materials are sufficient to illustrate conclusions made by the authors.*

*I congratulate the authors with such a nice piece of work.*

*I have just a few minor comments and suggestions and recommend publication after very minor changes.*

Thank you for your very kind words. We have addressed all your suggestions. Below you will find a detailed description of the changes made.

*Minor comments:*

1. *P.3, lines 15-20. Any way to merge these conflicting estimates?*

We agree that the presented estimates could be interpreted as conflicting. The estimates were at different depths and different locations, and so we clarified as follows.

15   fluxes of $\mathcal{O}(100)$ W m$^{-2}$ were measured (Sirevaag and Fer, 2009). Once the AW subducts, the vertical mixing is suppressed by the overlaying strong stratification, reducing the heat loss to the atmosphere or sea ice. Padman and Dillon (1991) observed a time-averaged upward heat flux in the pycnocline above the Atlantic layer of 25 W m$^{-2}$ over the YP slope, of that which, only about 6 W m$^{-2}$ actually entered the mixed layer. At the core of the Svalbard branch, Fer et al. (2010) observed that near-bottom mixing removed 15 W m$^{-2}$ from the AW layer to cold waters below. Outside the WSC, near the northeastern flank of the YP,

20   Sirevaag and Fer (2009) found an average vertical heat flux of 2 W m$^{-2}$, comparable to the annual oceanic heat flux of 3-4 W m$^{-2}$ to the Arctic pack ice (Krishfield and Perovich, 2005).

*2. P.5, Table 1: Can you please add mean vertical limits for the water masses?*

Thank you for this suggestion. In Table 1 we have inserted an extra column with the mean vertical limits, and revised the caption accordingly:

**Table 1.** Water masses as defined by Swift and Aagaard (1981) and Aagaard et al. (1985). Absolute Salinity ($S_A$) is calculated from the practical salinity ($S$) at 80°N and 10°E, and rounded to the nearest hundredth. The last column is the depth range of the different water masses observed in this study.

| Abbr. | Name | Practical Salinity $S$ | Absolute Salinity $S_A$ $(g\,kg^{-1})$ | Conservative Temperature $\Theta$ (°C) | Observed depth (m) |
|---|---|---|---|---|---|
| AW | Atlantic Water | $S > 34.88$ | $S_A > 35.05$ | $\Theta \geq 2$ | 45 - 475 |
| LAIW | Lower Arctic Intermediate Water | $S > 34.88$ | $S_A > 35.05$ | $2 > \Theta \geq 0$ | 475 - 790 |
| UAIW | Upper Arctic Intermediate Water | $34.88 \geq S \geq 34.7$ | $35.05 \geq S_A \geq 34.87$ | $\Theta < 1$ | not present |
| DW | Deep Water | $34.96 \geq S > 34.88$ | $35.13 \geq S_A > 35.05$ | $\Theta < 0$ | > 790 |
| PW | Polar Water | $S < 34.4$ | $S_A < 34.56$ | $\Theta < 0$ | 20 - 50 (Section A) |
| PIW | Polar Intermediate Water | $34.7 > S \geq 34.4$ | $34.87 > S_A \geq 34.56$ | $\Theta < 0$ | 50 - 65 (Section A) |
| ASW | Arctic Surface Water | $S < 37.4$ | $S_A < 34.87$ | $\Theta > 0$ | < 45 |
| | | $34.88 \geq S \geq 34.7$ | $35.05 \geq S_A \geq 34.87$ | $\Theta > 2$ | |

*3. P.10, line 29: Should the power be -2?*

The unit is correct. The heat loss presented by Boyd and D'Asaro (1994) is heat loss per downstream meter. This is now clarified.

*4. P.11, line 21: "Extracting" is not a clear word here.*

Agreed. We changed "extracting" to "reducing".

*5. P11, line 22-23: I do not see a point for placing this sentence ("Note that …") here.*

Agreed. We removed this sentence.

*6. P.11, line 30: Should it be comma instead of "-" after McPhee?*

The citation is correct. We are referring to an article by Erika E. McPhee-Shaw and Eric Kunze.

*7. P.12, line 35: How was the correspondence between buoyancy and heat fluxes found?*

It was found through the relationship $B = g\alpha F/(\rho_0 c_p)$, where B is the buoyancy flux, F is the heat flux and alpha is the thermal expansion coefficient of sea water. We now included this in the article.

25 $m^2\,s^{-1}$), an average geostrophic current of 0.3 $m\,s^{-1}$ and the observed range of across-slope bottom-density gradient, the lateral buoyancy flux is in the range between 0.3 and $4 \times 10^{-8}$ $W\,kg^{-1}$, sufficient to maintain the observed dissipation rates. When the salinity variations are negligible, the heat flux can be estimated as $\rho_0 C_P B/(g\alpha)$, where $\alpha$ is the thermal expansion coefficient of about $10^{-4}$ $K^{-1}$, and $B$ is the buoyancy flux. A buoyancy flux of $O(10^{-8})$ $W\,kg^{-1}$ corresponds to a turbulent heat flux of approximately 40 $W\,m^{-2}$.

When revisiting our calculation in response to the reviewer's comment, we realized a typo. The presented heat flux of 400 $W\,m^{-2}$ should be 40 $W\,m^{-2}$. We apologize for this huge error. We corrected it throughout. We also toned down our result and conclusion regarding the cooling induced by the

convectively-driven bottom boundary layer mixing ("can be important" instead of "at a rate comparable to that expected from diffusion by eddies"). The suggested mechanism is still a substantial contribution, larger than that expected from the interior diapycnal mixing.

In conclusions:

20    through a combination of mean shear and convection. Downslope Ekman advection across the slope leads to a lateral buoyancy flux of $O(10^{-8})\,\mathrm{W\,kg^{-1}}$, sufficient to maintain a large fraction of the observed dissipation rates, and corresponds to a heat flux of approximately $40\,\mathrm{W\,m^2}$. Convectively-driven bottom mixing can be important for cooling and freshening of the WSC.

8. *Fig. 10b: Any comment on elevated dissipation rate at ~400m in the third deepest profile (i.e. in the interior) and lack of such a signature in other profiles?*

The elevated dissipation rate at 400 m depth is presented in figure 1a below for reference. Turbulence is due to the high shear (fig. 1a) and the corresponding low Richardson number, *Ri* (fig. 1b). The same pattern is again seen at about 670 m depth where the shear is large and *Ri* is low. The lack of higher dissipation rate at 550 m depth where *Ri* is less than the critical value of 0.25, is likely due to the one-hour time difference between the current measurements and the microstructure measurements. The shear-induced turbulence within the ocean interior was beside the scope of this article, and therefore we do not include a new figure.

[Figure]

*Figure 1. Station A8, the third deepest profile. a) Absolute current velocity (blue) and dissipation rate of TKE (gray). b) Gradient Richardson number (Ri) with red and yellow lines showing Ri = 0.25 and Ri = 1 respectively.*

*The manuscript presents data from three shipboard sections occupied in summer 2015 northwest of Svalbard. The analysis focuses on explaining the along-pathway cooling of the West Spitsbergen Current concluding that vertical turbulent heat flux is not sufficient and that bottom mixing with shelf waters also makes a contribution. This material is relevant for a better understanding of the inflow of warm Atlantic Water to the Arctic Ocean and its modification processes along the inflow. This is appropriate and interest ing for the readership of Ocean Science. The manuscript is well written and presents the results in an easily digestible way even though the presented observations and analyses are complex. In my opinion, the authors missed one point in the discussion of their results which I would like the authors to comment upon/include in their discussion. Otherwise, the manuscript should be accepted after the correction of a few small suggestions. Therefore, I recommend minor revision.*

Thank you for your constructive suggestions. We revised the manuscript and especially improved our discussion. See below for a detailed description of the changes made.

*Main comment: Inherent to the your interpretation of the presented results are the assumptions that the current is uniform in the along pathway direction, that there is no temporal variability in the current, and that your observations sampled the current at the same time. This does not hold for two reasons: There is strong mesoscale variability and individual shipboard sections may even capture substantial southward flow in the WSC (e.g. Richter et al 2018 Ocean Science). Therefore, one would not necessarily expect volume conservation between consecutive (in along pathway distance) synoptic sections. Put another way, differences between the transport in consecutive sections do not need to correspond to volume transport loss from the current.*

True, we make the assumption that the conditions of the inner branch of the West Spitsbergen Current does not change significantly during the duration of this cruise. We now emphasize this point. Our four repeated stations, each lasting 12 hours or more, do not suggest significant changes, but variability on mesoscale can be substantial.

(SADCP). The sampling duration of sections A, B and C was approximately 20, 11 and 20 hours respectively. It took 5 days from the sampling started on section A to the end of C. Our results discussed in Sect. 4 assume that the conditions of the inner branch of the WSC do not change significantly during these 5 days, thus giving a synoptic view. In total, 46 CTD/LADCP and 85 VMP profiles are analyzed.

5    **2.1   Temperature and salinity measurements**

*Furthermore, there is a seasonal cycle in the temperature of the WSC and its extension. The maximum in temperature of the Atlantic Water (well below the highly stratified low salinity cap on top of the current) is reached later in the season north(east) of Svalbard (recent A-TWAIN results) compared to the WSC west of Svalbard. In my opinion this is mainly due to the northward advection of a characteristic seasonal cycle set further south. For example, consider that the seasonal cycle has a slope of 2.5°C in 5 month (=0.5°C/month) west of Svalbard (e.g. von Appen et al 2016 Journal of Physical Oceanography). If one – for the sake of argument – considers a mean advective velocity of 0.1m/s (also quoted on page 10 line 32), the advection from your section C to your section A (distance of 170km) would take approximately 19 days. In those 19 days, the current (due to the seasonal cycle) at the southern location would have warmed by approximately 0.3°C. Put another way, at any one point in time (prior to the seasonal maximum in AW temperature), one would expect it to be 0.3°C*

*cooler at section A than at section C. This corresponds to a horizontal temperature gradient of 0.18°C/100km, which is a number not substantially different from your 0.2°C/100km (page 10 line 19). On top of that comes that your different sections were obviously not occupied on the same day, but with some (consider specifying explicitly) days in between them. I do not want to claim that this explains everything which you see and your interpretation is probably still broadly appropriate for what is happening to the current. Nevertheless, I think it would be worthwhile to discuss these points and to carefully consider where they might (and where they would not) impact your conclusions.*

Thanks for bringing up this interesting point! We agree that this should be included in the discussion. The seasonal cycle of 2.5°C (that you mention) was measured at 75 m depth. This is close to the upper limit of our stream tubes, which span from about 50 to 500 m depth. von Appen et al. (2016) state that the seasonal cycle at 250 m depth is less than 1°C. Likely the seasonal cycle will decrease even more toward 500 m depth. It is more likely that an average seasonal cycle in our stream tubes is somewhere between 0.5 and 1°C. The mean velocity (0.1m/s) you refer to is what was assumed in other studies (clearly stated in text). The measured mean velocity in our study is 0.15 m/s, then it would take about 13 days to cover 170 km. In addition, it took 5 days from we started Section A until we finished Section C. Thus, over 18 days, the temperature change from the seasonal cycle would be between 0.06 to 0.12°C. The following discussion is amended in response to this comment:

An important point to consider when calculating the northward cooling rate is the effect of the seasonal temperature cycle. Using mooring observations in Fram Strait, von Appen et al. (2016) obtain a seasonal signal with temperatures increasing from April to September, with an amplitude of 2.5°C at 75 m, decreasing to less than 1°C at 250 m depth. The seasonal cycle is likely weaker below 250 m. Our stream tubes span from approximately 50 to 500 m depth. Assuming an average seasonal cycle in our stream tube temperature between 0.5°C and 1°C over 5 months (April to September), we can estimate the northward cooling rate expected from the seasonal cycle. Using our measured mean stream tube velocity of 0.15 m s$^{-1}$, a water parcel covers the 170 km distance from Section C to A in 13 days. In addition, we used 5 days from the start of Section A until we finished Section C, during which the temperature increase due to the seasonal cycle must be accounted for. Thus, over 18 days the average stream tube temperature in Section A would be between 0.06°C and 0.12°C less than in Section C, corresponding to a temperature loss of 0.04 to 0.07°C/100 km from the seasonal cycle alone. Moored observations also show that the rate of change of the seasonal signal is not constant, but weakens with time toward September when the temperature maximum occurs. A linear seasonal temperature gradient is likely not a good fit for the 18 days duration considered here, and a temperature loss in the lower range would be representative at the time of our cruise. Overall, we estimate approximately 0.05°C/100 km from the seasonal cycle, which is substantially less than the inferred cooling rate of 0.2°C/100 km.

*P1|11 Consider adding "We conclude that – at least in summer – convectively-driven bottom mixing…"*

Agreed. Changed as suggested.

*P1|12 Consider adding "can lead to substantial cooling and freshening of the WSC"*

Agreed. Changed as suggested.

*P2|21 Consider adding "These eddies may control"*

Agreed. Changed as suggested.

*P2|33 Note that these heat losses (in W/m^2) are dependent on the mean advective speed resp. the residence time of the water in the area of cooling.*

Agreed. We added the following at the end of the paragraph. P3, line 1

> 22-km wide core (Boyd and D'Asaro, 1994). Note that these heat losses are dependent on the mean advective speed, that is the residence time of the water in the area of cooling.

*P4|29 "averaging" Is this averaging in space or in time? Over what distance?*

It is averaging in time. The complete term defines the shear variance, and is obtained by integrating the shear wavenumber spectrum. We clarified this as follows:

> where $\nu$ is the kinematic viscosity, overbar denotes averaging in time, and the $\partial u/\partial z$ is the small scale shear of one horizontal velocity component $u$. Using a constant fall rate of the instrument and invoking the frozen turbulence hypothesis over an analysis time of several seconds, the term with the overbar represents the shear variance from order 1 m vertical scale to order 1 cm scales where dissipation occurs. Dissipation rates are calculated from the shear variance obtained by integrating the shear
> 5  wavenumber spectra, using 1-s FFT length and half-overlapping 4-s segments, following the corrections and methods described

*P5|21 Note that the smoothing does not necessarily remove all ageostrophic motion.*

We added:

> 25  (horizontal × vertical) window. While the smoothing removes the short time and length scale variability, it does not necessarily remove all ageostrophic variability.

*P5|28 Before this paragraph might be a good time to present the info on how long it took to occupy the individual sections. Is there e.g. contamination from multiple tidal cycles possibly represented incorrectly?*

Thank you for the suggestion. We now include this information under section 2, data. (P4, line 1.)

> (SADCP). The sampling duration of sections A, B and C was approximately 20, 11 and 20 hours respectively. It took 5 days from the sampling started on section A to the end of C. Our results discussed in Sect. 4 assume that the conditions of the inner branch of the WSC do not change significantly during these 5 days, thus giving a synoptic view. In total, 46 CTD/LADCP and

About the contamination from tidal cycles:
We use the inverse tidal model AOTIM-5 to remove tides from our current measurements. Sections A and C (lasting about 20 hours) might be affected by multiple tidal cycles, however, we assume detiding accounts for this. Furthermore, while the deep part of the sections (stations deeper than 750 m) take about 12 hours to complete, the core stations (and the shallow measurements) are taken relatively rapidly and hence are less affected by tidal variability. No action taken.

*P6|2 Here the consideration of my main comment come into play.*

We have amended the discussion in section 4.3 – cooling and freshening. The text from the manuscript is copied under your main comment above.

*P6|26 Does this not require that the transport in the stream is exactly constant, not just +-10%?*

Yes, it does. Unfortunately, using the gridded sections with 1 km horizontal resolution (section 3.2), we cannot exactly conserve the volume transport for each section. Also, the wider the streamtubes, the larger the transport uncertainties become considering that the LADCP error is ± 3 cm/s. This especially makes the uncertainties of stream tube in Section B large, because it is 61 km wide. This limitation is now clarified in section 3.3, P6:

> In the second alternative (stream tube 2), we conserve the volume flux (of the Svalbard branch) in the tube to within 10%
> 15  at all sections (dashed enclosed curves in Figs. 2, 3, and 4). Heat budget calculations (see Sect. 3.4) require conservation of volume. Given the 1-km horizontal resolution of our sections, and the measurement uncertainty in the velocity measurements, a conservation of volume to less than ±10% was not practically possible. The volume flux of 1.3 Sv at Section A is deemed representative of the Svalbard branch (see Sect. 4.2), and used as a constraint in Sections B and C. The lateral edges of the tube

*P7|27 e.g. Richter et al 2018 Ocean Science presents data from 2016 which is similar warm both near the surface and at greater depths as you describe.*

We added this information on P8, line 8-9:

> 5  northernmost and C is the southernmost section, and the horizontal distance is referenced to the location of the WSC core.
>     Temperatures near surface exceed 6°C in August, and decrease with depth in all three sections (Fig. 2). The northern part of Section A is close to the ice edge, and is characterized by cold surface waters. Compared to past observations, conditions in August 2015 were particularly warm, not only near the surface but also in the water column. Similar conditions were observed in summer 2016 by Richter et al. (2018). In October/November 2001 the AW temperatures above 4°C, west of Svalbard,

*P8|8 Consider "were higher in all sections by (range)." And then provide a range of salinity values.*

Revised as:

> 15  between 100 m and 400 m depth was found in all three sections, similar to previous studies in this region (Cokelet et al., 2008; Våge et al., 2016; Meyer et al., 2016). Compared to the climatology, the salinities were higher in all sections by 0.08-0.12 g kg$^{-1}$.

*P8|17-19 It might be helpful (though not necessary) to show a TS plot to better make this point.*

After consideration, we decided not to include this figure. Instead we refer to the open-access archived MSc thesis where this is shown:

> its depth-averaged temperature much faster than its salinity, implying mixing with colder water with similar salinity, such as LAIW (see also Kolås (2017) for $T - S$ diagrams).

*P8|24 At what x-value in your plots would the YP branch supposedly be if it were present?*

The Yermak branch supposedly follows the 1000 m isobath. That means it should be present somewhere between -5 and -15 km on the x-axis in section A and C. End of page 8, we inserted:

> along the 400-m isobath (Fig. 1b). The Yermak branch however, is not well-captured by the SADCP. If the Yermak branch was present in Sections A and C, we would expect to see evidence of this between $x = -5$ and $x = -15$ km. Over the YP, there

*P9|33 Consider "If … follows the f/H contours and there is no synoptic variability between the sections, …"*

Revised as suggested.

*P10|20 Be consistent with using "-0.20°C" versus "0.20°C" in this paragraph.*

In the manuscript we now use consistent figures throughout (from abstract to conclusions). For example:

> comparable to the downstream freshening of 0.013/100 km (in practical salinity scale) reported by Cokelet et al. (2008), and the 50-year mean summer freshening of 0.010/100 km, measured obtained by Saloranta and Haugan (2004). The northward temperature gradient corresponds to a cooling rate of 0.20°C/100 km for stream tube 1, and 0.23°C/100 km for stream tube
>
> 30    2. Cokelet et al. (2008) observed 0.19°C/100 km in fall 2001. Saloranta and Haugan (2004) observed a 50-year summer mean cooling rate of 0.20°C/100 km, same as that observed in 1910, between 75°N and 79°N (Helland-Hansen and Nansen, 1912).

*P11|3 "all vertical diffusive … (i.e. directed…)"*

Corrected as suggested.

*P11|6 "top", "bottom" It is not sure which values you refer to. Quote the numbers in your text.*

Agreed. (See also response to your last comment about Fig. 7.) We now changed the shaded bands indicating the top and bottom boundaries of the stream tube with dashed lines, and now quote the heat fluxes across these boundaries as follows:

> from top. Figures 7 and 8 show temporal mean of microstructure measurements at the repeated stations R1 and R4 respectively,
>
> 30    where the upper and lower dashed lines indicate the boundary of our stream tubes. The average profiles show a heat flux of $-5$ W m$^{-2}$ and $-4$ W m$^{-2}$ across the upper boundary of the stream tube in R1 and R4 respectively (Figs. 7e and 8e), whereas through the bottom boundary, the heat flux is only $-1$ W m$^{-2}$ (Fig. 7e). In Section A and C, where AW is above colder LAIW,

*P11|17 "If it is not cooled" Does it have to be cooled?*

Previous research and ours point to a northward cooling of the West Spitsbergen Current also during summer. It is unlikely that the northward temperature gradient is entirely a seasonal temperature change advected northward (see our response to your main comment). No action taken.

*P12|9 Consider "the qualitative pattern".*

Agreed. Changed as suggested.

*P12|14 "weakly-stratified"*

Done.

*P12|35 Vertical or horizontal heat flux.*

Clarified as "vertical heat flux"

*P13|2 "In winter…" Where does this supposedly (based on the reference) happen?*

Boyd and D'Asaro (1994) collected several sections across the WSC between 76.5 and 78 degrees north. The isopycnals were observed to outcrop 5-10km west of the core. Clarified on page 14:

> In winter, an energetic eddy field diffuses heat along steeply sloping, outcropping isopycnal surfaces, at a rate sufficient to cool
>
> 5    the subsurface warm core capped by stratification above (Boyd and D'Asaro, 1994). In winter, isopycnals in the core outcrop 5-10 km to the west of the core (Boyd and D'Asaro, 1994), whereas in our summer observations, the $\sigma_\theta = 27.7$ surface, an

*P13|18 Where/how was this average lateral temperature gradient estimated.*

The calculation ion P13 gives an order of magnitude estimate for the eddy fluxes. At each section, we calculate a lateral distribution of the vertically averaged AW temperature, then calculate the mean horizontal temperature gradient from this. We did this for all the sections. The given description is sufficient for the purpose. No action taken.

*P14|14 "less dense waters" Where do these waters come from? How is this pool sustained? You could either elaborate on this a bit in the introduction or (more valuable) you could use this information to do some speculations on how changes in the sources of these waters might affect the process that you looked at/the cooling to the WSC.*

Thanks this nice suggestion.  These waters are maintained by the coastal current, incorporating meltwater from glaciers, sea ice, and river runoff. Their properties would affect the resulting buoyancy flux. We now include a discussion of these shelf waters on page 13 (continued on page 14) as follows:

> 30     The source of buoyancy is the relatively less dense waters on the shelf, maintained by the West Spitsbergen Coastal Current. The coastal current is an extension of the East Spitsbergen Current, incorporating fresh and cold PW originating from the Arctic Ocean, melt water from glaciers, sea ice, and river runoff. While the shelf waters offer a sustainable pool of buoyant water along the path of the WSC, the changes in their temperature and salinity properties would affect the resulting buoyancy flux. Substantial variability of shelf water properties are reported on short-term, inter-annual and long-term scales in response
>
> to changes in large-scale atmospheric patterns (Goszczko et al., 2018). The consequences of the changes in the source shelf waters on the convectively-driven bottom boundary mixing and on the resulting cooling rate of the WSC merit further studies.

And also inserted a line on this in the conclusions:

> The WSC core on the slope flows in the direction of Kelvin wave propagation, inducing the downslope Ekman advection. Relatively less dense near-bottom waters on the shelf are the source of buoyancy, maintained by the cold and fresh waters from the Svalbard coast and fjords joining the west Spitsbergen coastal current. Our detailed observations show turbulence generation

*P15|29 This reference appears to be incomplete.*

Fixed.

*P18 LAIW line: The distinction between ">" and ">=" is meaningless for a continuous distribution.*

This distinction is consistent with previous definitions. No action taken.

*Fig2/3 The near surface stratification due to salinity is really hard to see in this (Fig2) kind of figure. Consider plotting salinity instead of Fig3. The benefits of Fig3 is not obvious.*

We experimented with plotting a figure similar to Fig. 2, but for salinity. The near-surface stratification due to salinity can only be shown by zooming in near the surface. Figure 3 shows the water masses, which also gives an idea of the stratification. The benefit of a figure showing only the surface stratification is not obvious to us (given the focus of our study, and with stream tubes extending from 75 m depth). No action taken.

*F6 Would this figure not be more effective with three subplots (1 for transport, 1 for temperature, 1 for salinity) and to have three differently colored lines in each of the subplots for each of the three sections.*

The presentation of section A, B and C as figure a, b and c is used throughout the article, and we prefer to maintain this consistency. Also, we point out that the vertically-integrated temperature maximum is located landward of the transport maximum, whereas the salinity maximum is not. This is more easily seen when transport, temperature and salinity is in the same subplot. No action taken.

*Fig7/Fig8 Is 700m equal to the bottom depth here? Consider to have the same spacing (in centimeters on the paper) of the y-axis in both Fig7 and Fig8.*

The profiles extend to the bottom. For Fig7 this is about 690 m, and for Fig8 this is about 475 m. The total depths are now given in the figure captions. The benefits of squeezing Fig8 so that the scaling of the y-axis is the same for both Fig7 and Fig8 is not obvious as these are separate figures. We prefer to have the figure tall enough to show the details.

*Fig7 caption: "Shaded bands" Why are these bands so wide?*

These bands were between two densities that made sure to capture the stream tube limit across the whole stream tube. This was not necessary for the temporal mean of the repeat stations R1 and R4. We now changed this shaded band into a dashed line that shows the stream tube limits. Revised figures:

[Figure]

**Figure 7.** Temporal mean of measurements collected by the microstructure profiler at station R1, Section A. (a) $\Theta$ (blue) and $S_A$ (red), (b) $\sigma_\theta$ (blue) and buoyancy frequency $N$ (red), (c) 15-m bin-averaged dissipation of TKE $\epsilon$, (d) eddy diffusivity $K_\rho$, and (e) vertical heat flux $F_H$. Dashed lines show the upper and lower boundaries of the stream tube. Water depth is 690 m.

[Figure]

**Figure 8.** Same as Fig. 7, but for station R4, Section C. Lower limit of the stream tube is the sea bed at 475 m.

------------- END of response to reviewers' comments. ----------------------

---

## Author Comment (AC2) · 18 Nov 2018

We thank reviewer 2 for her/his detailed review, constructive comments and useful suggestions. We have addressed all comments. Our detailed response can be found in the attached file which includes our response to both reviewers.

Please also note the supplement to this comment:
https://www.ocean-sci-discuss.net/os-2018-86/os-2018-86-AC2-supplement.pdf